# Electrically controlled nonvolatile switching of single-atom magnetism in a Dy@C$_{84}$ single-molecule transistor

Feng Wang[1,2,8], Wangqiang Shen[3,4,8], Yuan Shui[5,8], Jun Chen[1,2], Huaiqiang Wang[6], Rui Wang [1], Yuyuan Qin[1], Xuefeng Wang [7], Jianguo Wan[1], Minhao Zhang[1,2] ✉, Xing Lu [3] ✉, Tao Yang [5] ✉ & Fengqi Song [1,2] ✉

Single-atom magnetism switching is a key technique towards the ultimate data storage density of computer hard disks and has been conceptually realized by leveraging the spin bistability of a magnetic atom under a scanning tunnelling microscope. However, it has rarely been applied to solid-state transistors, an advancement that would be highly desirable for enabling various applications. Here, we demonstrate realization of the electrically controlled Zeeman effect in Dy@C$_{84}$ single-molecule transistors, thus revealing a transition in the magnetic moment from 3.8 $\mu_B$ to 5.1 $\mu_B$ for the ground-state G$_N$ at an electric field strength of 3−10 MV/cm. The consequent magnetoresistance significantly increases from 600% to 1100% at the resonant tunneling point. Density functional theory calculations further corroborate our realization of nonvolatile switching of single-atom magnetism, and the switching stability emanates from an energy barrier of 92 meV for atomic relaxation. These results highlight the potential of using endohedral metallofullerenes for high-temperature, high-stability, high-speed, and compact single-atom magnetic data storage.

As scaling the dimensions of magnetic media materials toward the spatial limit, magnetic single atoms are considered the ultimate goal of such downsizing[1–5], which have thus been an active field of research. However, realizing single-atom magnetic data storage poses significant challenges, including the preservation of unstable single-atom magnetism for prolonged manipulation times. Studies on magnetic single atoms have demonstrated magnetic remanence and relatively high magnetic stability of Ho and Dy atoms for reading and writing on decoupled thin insulating layers[2–7]. The bistability of atomic magnetism significantly depends on the symmetry-protected magnetic ground state placed within a carefully designed coordination field[2,6–9]. Additionally, despite the large magnetic anisotropy energies (MAEs) exhibited by individual atoms, the occurrence of quantum tunneling of magnetization (QTM) reduces the energy barrier[2,3,5], which results in the loss of useful magnetism and affects the stability of the magnetic state.

[1]National Laboratory of Solid State Microstructures, Collaborative Innovation Center of Advanced Microstructures, School of Physics, Nanjing University, Nanjing 210093, China. [2]Institute of Atom Manufacturing, Nanjing University, Suzhou 215163, China. [3]State Key Laboratory of Materials Processing and Die & Mould Technology, School of Materials Science and Engineering, Huazhong University of Science and Technology, Wuhan 430074, China. [4]School of Materials Science and Engineering, Hefei University of Technology, Hefei 230009, China. [5]MOE Key Laboratory for Non-Equilibrium Synthesis and Modulation of Condensed Matter, School of Physics, Xi'an Jiaotong University, Xi'an 710049, China. [6]Center for Quantum Transport and Thermal Energy Science, School of Physics and Technology, Nanjing Normal University, Nanjing 210023, China. [7]State Key Laboratory of Spintronics Devices and Technologies, School of Electronic Science and Engineering, and Collaborative Innovation Center of Advanced Microstructures, Nanjing University, Nanjing 210023, China. [8]These authors contributed equally: Feng Wang, Wangqiang Shen, Yuan Shui. ✉e-mail: zhangminhao@nju.edu.cn; lux@hust.edu.cn; taoyang1@xjtu.edu.cn; songfengqi@nju.edu.cn

In addition, demonstrating single-atom magnetic data storage in solid-state transistors remains challenging. Attempts at single-molecule transistors (SMTs) based on single magnetic atoms coupled with ligands have revealed electronic spin states, magnetic excited states, and atom–ligand exchange coupling; nevertheless, information on switching atomic magnetism has rarely been obtained[10–16]. Furthermore, in many experiments, the manipulation of magnetic states depends on magnetic fields[2,4–6], which hinders the realization of rapid, spatially compact control. An electric field has previously been shown to manipulate the geometrical state of endohedral metallofullerene (EMF), resulting in the transition of two bistable states with different permanent electric dipole orientations[17,18]. If it is possible to manipulate the coordination around the magnetic atom in EMFs through an electrical field, the resulting magnetic moment should differ, and switching of single-atom magnetism can theoretically be realized through the magnetoelectric coupling (MEC) effect[19–25].

Here, we present the successful switching of single-atom magnetism in Dy@$C_{84}$ SMTs for nonvolatile data storage. The two electrically controlled molecular states exhibit distinct magnetic moments, as demonstrated by the Zeeman effect. In addition, they display different magnetoresistance (MR) ratios at the resonant tunneling point with values of 600% for molecular state 1 and 1100% for molecular state 2, respectively. Density functional theory (DFT) calculations further confirmed the possibility of switching from one molecular state to another using an electrostatic field, and the coordination around the magnetic Dy atom changed. Therefore, the magnetic moment can be expected to be different to realize magnetism switching.

## Results

### Device fabrication and charge transport measurements

We used the precious trivalent EMF Dy@$C_{84}$ (see the right panel of Fig. 1a); its synthesis and purification processes are provided in the Methods section and Ref. 26. The internal Dy atom donates one $4f$ and two $6s$ electrons to the outer cage[26], which decreases the shielding effect of the cage through hybridization and brings the $4f$ orbitals of the Dy atom closer to the Fermi level for enhancing the electrical access and manipulation capabilities[27,28]. Three-terminal Dy@$C_{84}$ SMTs were fabricated to allow reliable modulation of the molecular chemical potential by a back gate, as depicted in Fig. 1a. A Dy@$C_{84}$ molecule bridges a pair of Au electrodes created by the feedback-controlled electromigration break junction (FCEBJ) technique, and electrical transport through the Dy@$C_{84}$ SMTs was investigated at a cryogenic temperature of 1.8 K (see the Methods section and Refs. 17,29. for more details).

Figure 1b shows the source-drain current vs. the source-drain voltage ($I_{sd}$-$V_{sd}$) traces of the FCEBJ process. Representative $I_{sd}(V_{sd})$ curves measured at different gate voltages ($V_g$) reveal apparent non-linear behaviors and Coulomb blockade in the low bias region (within $V_{sd}$ ~ 20 mV), as shown in Fig. 1c. We can thus access the single-electron tunnelling regime, in which the source-drain current exhibits Coulomb resonances as a function of $V_g$ when $V_{sd}$ is held constant ($V_{sd} = 5$ mV) (see Fig. 1d). Additionally, upon sweeping $V_g$ forward and backwards over a wide range of up to ±10 V, we observed two distinct sets of Coulomb oscillation patterns. Another device (device B) exhibited the same Coulomb blockade effect, as shown in Supplementary Fig. 1.

### Electrically controlled Zeeman effect in Dy@$C_{84}$ SMTs and the transition of the magnetic moment

The magnetic properties of Dy@$C_{84}$ SMTs were investigated by tunneling spectroscopy under magnetic fields via the Zeeman effect. We focus on the charge degeneracy point dominated by the resonant tunneling of an electron between the $N-1$ and $N$ charge states through an individual molecule[10,15] ($V_g = -7.2$ V and $V_g = -7.9$ V, marked as red and blue arrows in Fig. 1d, respectively), and then plot coloured maps of the differential conductance ($dI_{sd}/dV_{sd}$) around each peak with respect to $V_{sd}$ and $V_g$ in Fig. 2a, d, which are obtained through

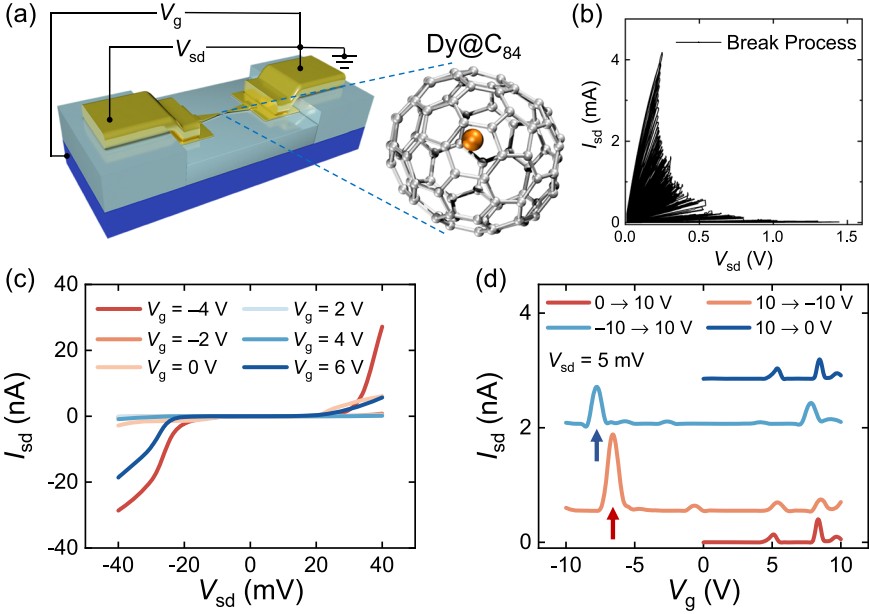

**Fig. 1 | Intrinsic molecular design and electron transport of the Dy@$C_{84}$ single-molecule transistor (SMT). a** Schematic of a three-terminal Dy@$C_{84}$ SMT and the Dy@$C_{84}$ molecule. A Dy@$C_{84}$ molecule falls into the nanogap of ~1 nm created by a feedback-controlled electromigration break junction (FCEBJ) process and forms a Dy@$C_{84}$ SMT. **b** Typical source-drain current vs. source-drain voltage ($I_{sd}$-$V_{sd}$) traces of the FCEBJ process. **c** Representative $I_{sd}(V_{sd})$ characteristic curves at different gate voltages ($V_g$) after electromigration exhibiting Coulomb blockade in the low bias region (within $V_{sd}$ ~ 20 mV) ($T = 1.8$ K; all transport measurements were performed at this cryogenic temperature). **d** The source-drain current $I_{sd}$ recorded with respect to $V_g$ with a fixed $V_{sd}$ of 5 mV when we sweep $V_g$ back and forth within the range of ±10 V. The curves have been offset vertically for clarity. Two sets of Coulomb oscillations emerge, and we focus on the resonant tunneling points marked by the red and blue arrows.

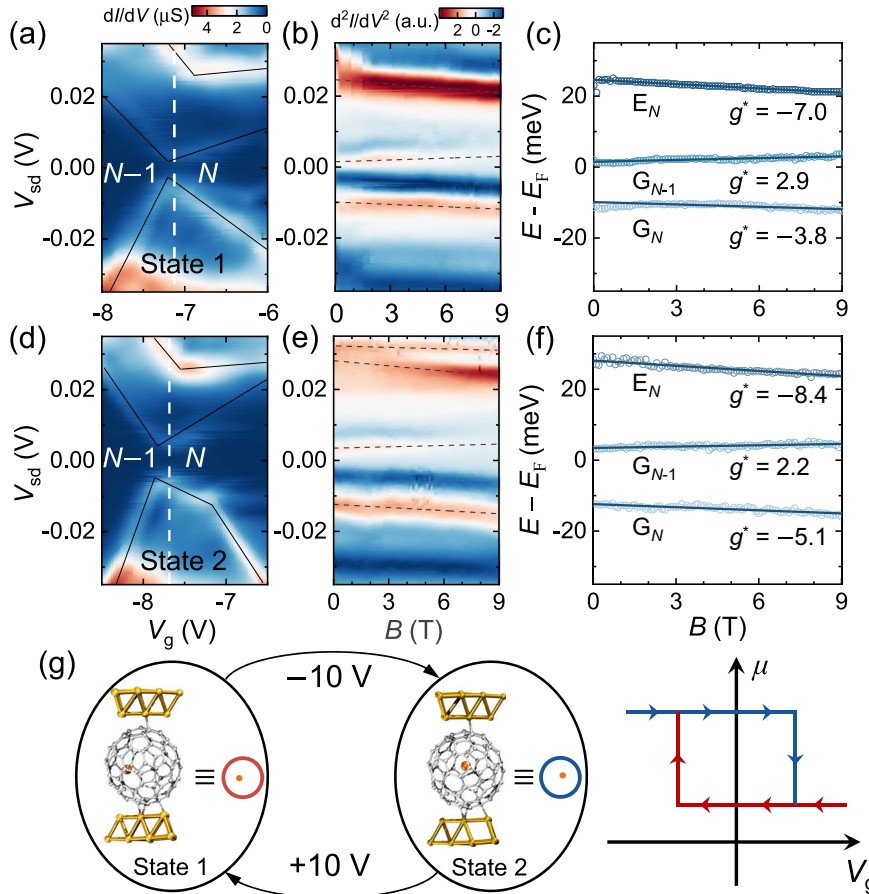

**Fig. 2 | Electrically controlled nonvolatile switching of single-atom magnetism. a, d** Colored maps of the differential conductance (d$I$/d$V$) near the charge degeneracy point for the transition between $N-1$ and $N$ electrons ($V_g = -7.2$ V and $V_g = -7.9$ V marked as red and blue arrows in Fig. 1d, respectively) for state 1 (**a**) and state 2 (**d**) under a zero magnetic field ($B = 0$ T). The black lines indicate the main resonant excitations of the ground and excited states. Colored maps of d$^2I_{sd}$/d$V^2_{sd}$ as a function of $B$ and $V_{sd}$ for state 1 (**b**) and state 2 (**e**) at constant $V_g$ ($V_g = -7.1$ V for Fig. 1b and $-7.7$ V for Fig. 1e and marked by the white dashed lines parallel to the vertical axis in Fig. 2a, d). The d$^2I_{sd}$/d$V^2_{sd}$ maps provide clearer indications of energy shifts in ground and excited states with $B$. The excited state of state 2 exhibits a clear split-like behavior, likely attributed to the crossover between the $N-1$ and $N$ excited states rather than real energy splitting under magnetic fields. **c, f** The

relative energies of the ground and excited states of state 1 and state 2 plotted as a function of $B$; these energies are extracted from the positions of the d$^2I_{sd}$/d$V^2_{sd}$ peaks in (**b**) and (**e**). We then fit the data linearly according to the Zeeman effect, and the effective $g$-factor was defined as: $g^* = \frac{\Delta E}{\mu_B \Delta B}$. The values of $g^*$ are shown in the figure. These states with measured $g^*$ values are attributed to mixed states of Dy$^{3+}$ $m_J$ states or more likely metal-cage hybrid states. **g** Switching operation of the molecular states in the Dy@C$_{84}$ SMT. The molecular states represented by two different molecular conformations can be switched by a gate voltage of ±10 V. The right panel shows schematic of a typical electrically controlled nonvolatile switching of single-atom magnetic moment corresponding to the molecular state transition.

## Table 1 | Effective $g$-factors and magnetic moments extracted from the tunneling spectra in each molecular state

| Molecular state | | Effective $g$-factor ($g^*$) derived from d$^2I$/d$V^2$ peaks | Effective magnetic moment ($\mu_B$) |
|---|---|---|---|
| State 1 | ES($N$) | $-7.0 \pm 0.1$ | 7.0 |
| | GS($N-1$) | $2.9 \pm 0.2$ | 2.9 |
| | GS($N$) | $-3.8 \pm 0.3$ | 3.8 |
| State 2 | ES($N$) | $-8.4 \pm 0.4$ | 8.4 |
| | GS($N-1$) | $2.2 \pm 0.2$ | 2.2 |
| | GS($N$) | $-5.1 \pm 0.3$ | 5.1 |

numerical differentiation of the current. The Coulomb diamonds exhibit the $N-1$ and $N$ ground states, as well as an excited state (~ 30 meV) parallel to the $N$ ground state. To investigate the response of ground and excited states to an external magnetic field ($B$), $I_{sd}(V_{sd})$ curves are recorded while sweeping $B$ perpendicular to the current direction at a constant $V_g$ ($V_g = -7.1$ V and $V_g = -7.7$ V, as shown by the

white dashed lines in Fig. 2a and d, respectively). Figure 2b, e shows 2D maps of d$^2I_{sd}$/d$V^2_{sd}$ as a function of $V_{sd}$ and $B$, which offer clearer indications of level shifts. In particular, the excited state in Fig. 2e exhibits an apparent split-like behaviour with $B$, which is attributed to the crossover between the $N-1$ and $N$ excited states (rather than the effects of real energy splitting). The relative energies of the ground and excited states extracted from the peaks position of the d$^2I_{sd}$/d$V^2_{sd}$ curves at different $V_g$ in Fig. 2b, e are shown in Fig. 2c, f. As $B$ increases, an upwards shift in the relative energy levels of G$_{N-1}$ ground states occurs, with a corresponding downwards shift in the G$_N$ ground states and E$_N$ excited states. However, these measurement results of this device do not exhibit energy splitting, indicating that no Kramers doublet is present in these ground states[30,31].

By applying a linear fit to the level shift according to the Zeeman effect, it is possible to determine the effective $g$-factor[32,33], which is defined as $g^* = \frac{1}{\mu_B} \frac{\Delta E}{\Delta B}$ (where $\mu_B$ is the Bohr magneton). The absolute values of $g^*$ range from 2.2 to 8.4 (Table 1). In comparison with the observed $g \approx 2$ for a C$_{60}$ molecule[34,35], the higher absolute values of $g^*$ can be qualitatively attributed to the contribution of the orbitals of Dy$^{3+}$. As the ground state $|J = 15/2\rangle$ of free Dy$^{3+}$ leads to a $g^*$ value of -10

(Supplementary Note 1), these ground and excited states with $g^*$ from 2.2 to 8.4 occur due to either the mixing of different $m_J$ states or more likely hybridization between orbitals of the carbon cage and the $Dy^{3+}$ ion, which is susceptible to the coordination environment. Moreover, there are disparities in $g^*$ (and thus disparities in the effective magnetic moments) between the two states in Fig. 2a and d, which can be switched by reversible electrical control (see Table 1). Specifically, the magnetic moments of the ground-state $G_N$ are 3.8 $\mu_B$ and 5.1 $\mu_B$, respectively. We demonstrate that these disparities in $g^*$ between the two molecular states are likely related to changes in the coordination environment and thus the metal-cage hybrid orbitals in the DFT calculations below.

Consequently, we designate the state in Fig. 2a (red line in Fig. 1d) as molecular state 1 (state 1 for short) and the alternative state in Fig. 2d (blue line in Fig. 1d) as state 2, demonstrating an electrical switching operation between two molecular states and, consequently, the single-atom magnetic moment of $Dy^{3+}$ as shown in Fig. 2g. The right panel illustrates a schematic hysteresis loop that essentially represent a prototypical electrically controlled nonvolatile switching of single-atom magnetism. Considering a silicon oxide layer thickness of ~10−30 nm and a switch gate voltage of 10 V, the variation in magnetic moments ($\Delta\mu$) for the $G_N$ ground state is approximately 1.3 $\mu_B$ at an electric field strength of ~3−10 MV/cm. These advancements make corresponding data storage applications more feasible.

### Large MR and potential multistate data storage

In this section, we demonstrate the disparity in MR between the two molecular states to highlight its potential for magnetic data storage. We investigated the evolution of the $N−1/N$ resonance peaks (marked as red and blue arrows in Fig. 1d) with respect to an external magnetic field. In Fig. 3a, b, with a fixed $V_{sd}$ of 5 mV, $I_{sd}$-$V_g$ curves around the $N−1/N$ resonant tunneling point (in Fig. 2a, d) of the two molecular states are plotted at various magnetic fields, revealing a significant reduction in the current intensity as $B$ increases. The current intensities of the two bistable states exhibit distinct responses to magnetic fields, highlighting the versatility of this approach for multistate data storage applications. For example, this approach enables the realization of eight current intensity values from two states, denoted as 11 to 24 in Fig. 3c, which serve as distinct data storage states and facilitate electrical writing through electric field switching of molecular states and magnetic reading by detecting distinct MR using a magnetic field.

The corresponding ratios of MR are calculated ($MR = (R(B) − R(0))/R(0)*100\%$)) and plotted as a function of $B$ (see inserts in Fig. 3a, b). In addition to exhibiting high MR ratios of up to 600% for state 1 and 1100% for state 2 at a magnetic field strength of 9 T, the manipulation of molecular states enables MR switching by applying a threshold gate electric field. The MR of state 2 is -180% larger than that of state 1 under the same magnetic field. Notably, the observed MR of up to 1100% is remarkably high and remains unaffected by the shift in peak position due to the Zeeman effect. We compare the MR of the $Dy@C_{84}$ SMT with those observed for other molecular devices[36–42] in Fig. 3d, which shows that our device outperforms other molecular systems by nearly an order of magnitude in terms of MR.

Considering the mechanism of MR, we exclude the possibility of the molecule–electrode interface mechanism being reported in nonresonant tunnelling regions of some organic radicals[36–40]. Furthermore, the effect of spin polarization reported in negative MR systems[42]

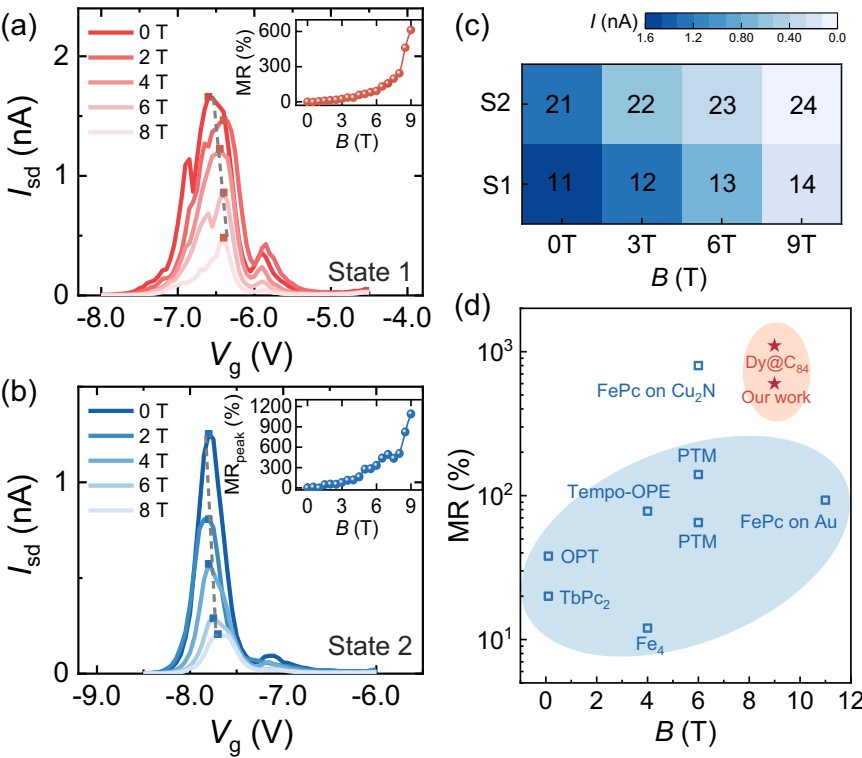

**Fig. 3 | Large MR and multistate operation of $Dy@C_{84}$ SMTs. a, b** Evolution of the $N/N−1$ resonant tunneling peaks (marked as red and blue arrows in Fig. 1d) in state 1 (**a**) and state 2 (**b**) with respect to magnetic field $B$ when fixing $V_{sd} = 5$ mV. The current intensities of these resonance peaks are suppressed by the magnetic field. The inserts show the magnetic field dependence of MR extracted from peak amplitudes of the $N/N−1$ resonant tunneling point for state 1 and state 2, indicating high MR ratios of up to 600% for state 1 and 1100% for state 2 at 9 T. **c** Comparison of the amplitudes of current peaks (labelled 11 to 14 and 21 to 24) in state 1 (S1) and state 2 (S2) under varying magnetic fields ($B = 0, 3, 6, 9$ T) extracted from (**a**) and (**b**). This approach can be used to achieve multistate data storage. (**d**) Summary of MR values obtained from different molecular devices, namely, FePc (Refs. 36,37), PTM (Ref. 38), Tempo-OPE (Ref. 39), OPT (Ref. 40), Fe4 (Ref. 41), and TbPc2 (Ref. 42), and our device over a range of $B$. The MR of our study exceeds those of other atomic or molecular devices by nearly an order of magnitude.

is also unreasonable for explaining the positive values here. The resulting MR behaviours observed in our device can thus be explained by the metal-cage hybrid state, as discussed in the following section.

### DFT calculations further corroborate the electrically nonvolatile switching of single-atom magnetism

The spin–orbit interaction effect is generally negligible for atomic-scale electron transport, which makes difficult the electrical manipulation of single-atom magnetism without magnetic fields[20,43–46]. Additional factors such as electrical modulation of exchange interactions and broken symmetries associated with electric dipole moments have recently been employed for spin–electric control[20,23,25,47]. The transition of the encapsulated atom between two different sites in certain EMFs with intrinsic broken inversion symmetries results in two states with different permanent electric dipole orientations[17,18], which can be utilized for reversible electrical manipulation of the magnetism of the encapsulated atom.

However, a carbon cage can block the effect of an electric field[28]. It is therefore somewhat difficult to affect an internal atom. Hybridization between the Dy and cage orbitals decreases the shielding effect of the cage. Previous report has also indicated that the accessible orbitals of Dy@C$_{82}$ EMFs, when deposited on a Ag surface, are primarily contributed by the 4$f$ orbitals of the Dy atom and the metal-cage hybrid orbitals[27]. Our DFT calculations show that the HOMO, LUMO, and their adjacent molecular orbitals likely originate from either the 4$f$ orbitals of Dy$^{3+}$ or the metal-cage hybrid orbitals (Fig. 4a, c and Supplementary Fig. 5, Supplementary Fig. 6, and Supplementary Table 1).

The mechanism of electrically controlled nonvolatile switching between two molecular states is qualitatively validated with theoretical calculations. The bistable molecular states are identified as the two most stable molecular conformations. State 1 is approximately 60 meV more stable than state 2 in free neutral Dy@C$_{84}$ molecules, while the energy barrier of bistable switching is -145 meV (see Supplementary Fig. 7 and Supplementary Table 2). The influence of the Au electrodes

on molecules was further investigated by considering the coupling between the electrodes and molecules. In this configuration, the energy difference between two molecular states is reduced to 2 meV, while the energy barrier decreases to 92 meV. The charge transfer with the electrodes also influences the energy difference and energy barrier between the bistable states (Supplementary Table 2). As shown in Fig. 4b, the gate electric field effectively reduces the energy barrier to a negligible level when the electric field reaches 0.25 V/Å, leading to a molecular state transition. This transition is accompanied by a displacement of the Dy atom, a modification to the coordination environment, a rearrangement of molecular orbitals and density of states (DOSs), and charge redistribution (Fig. 4a, c, d). Consequently, there is a significant change in orbital angular momentum, which affects the magnetic moment. Specifically, the magnetic moment of the Dy atom switches from 4.4 $\mu_B$ in state 1 to 5.0 $\mu_B$ in state 2 (Table 2). The magnetic moment of the Dy@C$_{84}$ EMF is mainly contributed by the Dy atom according to the calculation results, and the experimentally observed large g* values arise from the metal-cage hybrid states, which are sensitive to the coordination environment.

Furthermore, the observed significant MR and the variation in MR between the two molecular states are attributed to the effect of the metal-cage hybrid state. Due to the difference in effective g-factors between the Dy and cage components, an increase in the magnetic field leads to an apparent decrease in the degree of orbital overlap and the density of the hybrid states. For instance, when $\triangle g^* = 5$, a magnetic field of 9 T induces an overlap gap of almost 3 meV between Dy and cage components. According to the Landauer model, a decrease in the DOS of the hybrid state would lead to a drastic reduction in the current intensity, as shown in Fig. 3a, b. Furthermore, the degree of hybridization may influence the MR properties. The smaller MR and variation values may be attributed to a low degree of hybridization (device B in Supplementary Figs. 2 and 3).

In addition to their distinct magnetic moments, our calculations reveal that both molecular states exhibit unique electric dipole

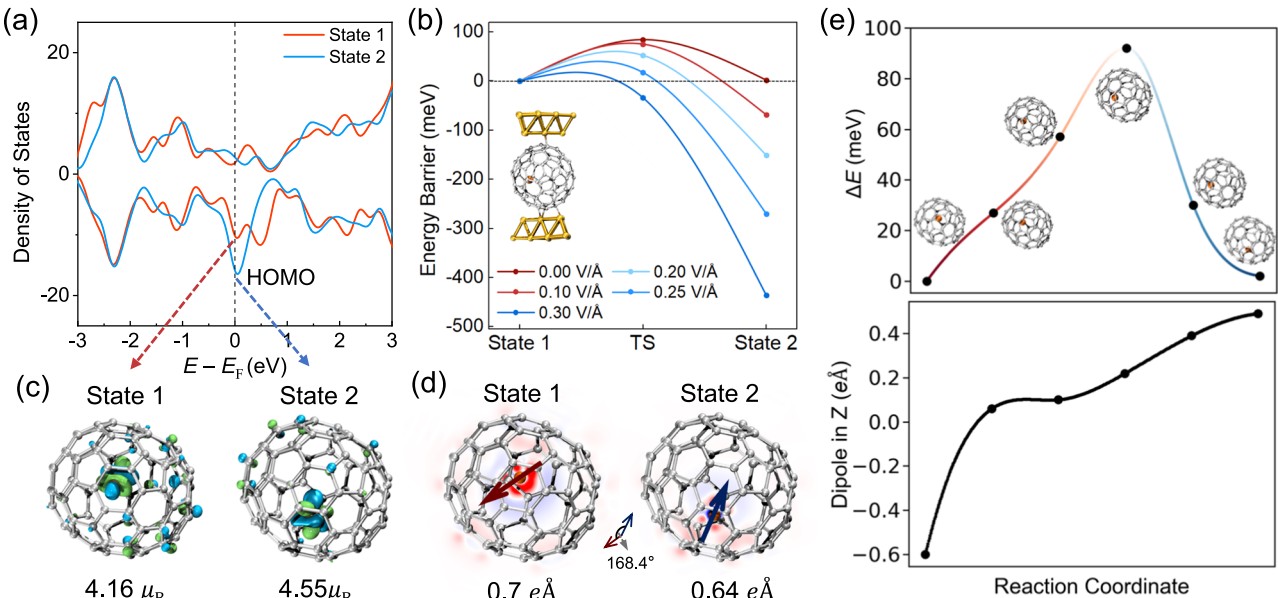

**Fig. 4 | DFT calculations for the two molecular states. a** Calculated density of states (DOSs) of the two molecular states for the free neutral Dy@C$_{84}$ molecule in the near range of the Fermi level. The Fermi level is shifted to the HOMO. **b** Calculated energy barriers between two molecular states across the transition state (TS) under various gate electric fields. The gate electric field can effectively lower the energy barrier to a negligible level at -0.25 V/A, thereby enhancing the transition probability between the two molecular states. The insert shows the interaction between the molecule and the Au$_{16}$ electrode. **c** Two possible orbital configurations of metal-cage hybrid states in two molecular states at the HOMO, with magnetic moments 4.16 $\mu_B$ and 4.55 $\mu_B$. **d** Upon switching the molecular states, the coupled magnetic moment of the ground state transforms from 4.16 $\mu_B$ for state 1 to 4.55 $\mu_B$ for state 2, while the electric dipole moment transforms from 0.70 $e$Å to 0.64 $e$Å. The two electric dipoles exhibit a relative angle of 168.4°. **e** The energy profile with structural diagrams of the Dy ion position (the upper panel) and the dipole moment in the z-direction (the lower panel) when climbing the energy barrier.

**Table 2 | Calculated relative energies, electric dipole moments, and effective magnetic moments of state 1 and state 2**

| Molecular state | Energy (free molecules) (meV) | Electric dipole moment (eÅ) | Effective magnetic moment of Dy@$C_{84}$ | Effective magnetic moment of Dy |
|---|---|---|---|---|
| State 1 | 0 | 0.70 | 4.16 $\mu_B$ | 4.4 $\mu_B$ |
| State 2 | 60 | 0.64 | 4.55 $\mu_B$ | 5.0 $\mu_B$ |

moments (Table 2 and Fig. 4d). When switching the molecular states using an electric field, the coupled magnetic moment of the ground state transforms from 4.16 $\mu_B$ for state 1 to 4.55 $\mu_B$ for state 2, while the electric dipole moment transforms from 0.70 $e$Å to 0.64 $e$Å. These two electric dipoles have a relative angle of 168.4°. The energy profile depicting structural diagrams of the molecule (the upper panel in Fig. 4e) and the dipole moment in the z-direction (the lower panel in Fig. 4e) climbing along the energy barrier are investigated to provide comprehensive insights into the bistable state transition. Therefore, as an example of a lanthanide EMF, Dy@$C_{84}$ demonstrates considerable promise as an innovative platform for showcasing MECs.

## Discussion

In summary, we have successfully achieved electrical tuning of the atomic magnetic moment in Dy@$C_{84}$ SMTs, providing a novel approach to realize single-atom magnetic data storage. Our results demonstrate that by applying a gate electric field of ~3−10 MV/cm, two bistable molecular states can be switched. These bistable molecular states exhibit a significant Zeeman effect with large effective $g$-factors. Specifically, the magnetic moment of the ground-state $G_N$ transformed from 3.8 $\mu_B$ to 5.1 $\mu_B$. Moreover, the MR of the $N{-}1/N$ resonant tunneling point respectively transitioned from 600% in state 1 to 1100% in state 2. Based on metal-cage hybridization, we manipulate the magnetic properties of the Dy atom within solid-state transistors, paving the way for advances in energy-efficient magnetic data storage and electronics with high integration and low power consumption.

## Methods

### Synthesis and isolation of Dy@$C_{84}$

Raw soot containing dysprosium EMFs was synthesized using a direct-current arc discharge method[48]. The graphite rods were packed with a mixture of $Dy_2O_3$ and graphite powder (molar ratio of Dy/C = 1:12) and annealed at 1000 °C for 7 h under an argon atmosphere. Then, these graphite rods were vaporized in an arcing reactor under a 300 Torr helium atmosphere with an arc current of 100 A. The pure Dy@$C_{84}$ compound was finally obtained through a combination of Lewis acid treatment and high-performance liquid chromatography (HPLC) separation[26].

### Device fabrication

Three-terminal transistors were fabricated using standard nanofabrication techniques, including UV lithography, electron beam lithography (EBL), and electron beam evaporation (EBE). The gate and external circuitry were defined by UV lithography, while the source and drain electrodes were patterned using EBL. A 30-nm layer of silicon oxide grown by atomic layer deposition (ALD) was used as the dielectric layer of the backgate, on which gold (Au) nanowires (width approximately 50 nm) were deposited by EBE as source and drain electrodes. After cleaning the nanowire transistors using oxygen plasma, a dilute drop of Dy@$C_{84}$ solution (0.1 mmol/L) was deposited on the device before drying. The device was then cooled to a cryogenic temperature of 1.8 K (all measurements were performed at 1.8 K), and nanogaps were formed using the FCEBJ method by monitoring the current through the nanowire while increasing the applied bias voltage on the source and drain electrodes. The cycle was repeated by increasing the applied voltage from 0 mV again once the current had

decreased by 1%, until the conductance reached 0.02 $G_0$ at a voltage of 20 mV (where $G_0 = e^2/h$). The current data from the transport measurements were smoothed, and the original data are listed in Supplementary Fig. 4. The molecule-Au couplings were typically greater than the typical intramolecular exchange (~1 meV)[12]. Therefore, the intramolecular exchange-induced effects could not be resolved in the transport spectra.

### Theoretical calculations

Geometrical optimizations of Dy@$C_{84}$ were performed using DFT calculations implemented in the program package ADF2019[49]. The Perdew–Burke–Ernzerhof (PBE) functional[50] with D3 dispersion correction[51] and TZ2P basis set were employed with the scalar relativistic zeroth-order regular approximation (ZORA)[52] Hamiltonian for relativistic effects. The frontier molecular orbitals (FMOs) and density of states (DOS) were plotted at the same level of theory with a Gaussian smearing width of 0.3 eV. The magnetic moment was calculated with the Vienna ab initio simulation package (VASP)[53] using the PBE-D3 functional. The interaction between valence electrons and ionic cores was considered within the framework of the projector augmented wave (PAW) method[54,55]. The energy cut-off for the plane wave basis expansion was set to 400 eV, while the criterion for total energy convergence was set at $10^{-5}$ eV.

## Data availability

All data supporting the findings of this study are available within the main text and the Supplementary Information file. The data that support the findings of this study are available from the corresponding authors upon reasonable request. Source data are provided with this paper.

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

## Acknowledgements

We acknowledge the financial support of the National Key R&D Program of China (Grant No. 2022YFA1402404), the National Natural Science Foundation of China (Grant Nos. 92161201, T2221003, 21925104, 22001084, 92261204, 12104221, 12104220, 91961101, 12025404, 62274085, T2394473, T2394470, 61822403, 12104217, and 12274337), and the Fundamental Research Funds for the Central Universities (Grant No. 020414380192).

## Author contributions

F.S. and M.Z. conceived the research. X.L. and T.Y. co-supervised the project. F.W. performed the device fabrication and experimental measurements. W.S. and X.L. prepared the molecular materials. Y.S. and T.Y. performed the density functional theory calculations. F.W. and M.Z. analyzed the data presented in the paper and Supplementary Information. F.W., M.Z. and F.S. wrote the manuscript. W.S., Y.S., J.C., H.W., R.W., Y.Q., X.W. and J.W. participated in discussions on this manuscript. All authors discussed the results and commented on the manuscript.

## Competing interests

The authors declare no competing interests.
