## [Peer Review File · Nature Communications]

Electrically controlled nonvolatile switching of single-atom magnetism in a Dy@C84 single-molecule transistorREVIEWER COMMENTS

Reviewer #1 (Remarks to the Author):

The paper 'Electrically controlled non-volatile switching of single-atom magnetism in a Dy@C84 single-molecule transistor' address an interesting and forefront subject of research, however it is so badly written and as consequence it is difficult if not impossible to review. This can be noticed since the incipit. In the abstract, the Authors state 'The atomic manipulation and manufacturing have for some time been transformative, and represented the frontier of research in the nanotechnology of high-density storage.'

For example they do not specify to which storage they refer to, presumably they refer to data storage.

The experimental conditions under which the data reported in Figure 2 (a and d) are acquired are not clear at all and this renders rather difficult to follow the rest of the manuscript.

I suggest a careful rewriting of the manuscript before it can undergo to a review process.

Reviewer #2 (Remarks to the Author):

The result is highly interesting and in my opinion deserves publication in Nature Communications. That being said, let me offer a couple of comments and questions:

1: I think primarily referring to the two states in the system as "two magnetic states" is dangerously misleading and should be avoided. While indeed the two states present a different magnetic moment, and while the two states present different values of magnetoresistance, the nature of the two-level-system (TLS) is fundamentally not magnetic, rather geometrical: the Dy is in either of two possible positions in space. Naturally, this affects the electric dipole both in direction and in magnitude, and also the coordination around the Dy, so in principle also the magnetic dipole can be expected to be different.

Elaborating further:

1a) DFT calculations could show the geometrical "reaction path" with the lowest activation energy between "reactives" (state 1) and "products" (state 2): how does the Dy ion move? This would basically translate into improving the information provided in the current version of Fig 4b.

1b) Movement of a charge between two positions would primarily change the direction of the electric dipole and only secondarily its magnitude. This is represented on Figure 4f, but we could benefit from a focus on the direction, e.g. the relative angle between the two dipoles, or the two relative angles between each of the electric dipoles and the external electrical field, if it can be estimated.

1c) In each of the two geometrical states there is a very strong MR, in the order of 10^3 . This may be interesting in itself... if the spin dynamics between each of the two magnetic states within each of the two geometrical states was slow enough to employ it as an additional memory bit. However, if one is just employing the geometrical TLS for memory and the MR as a way of measuring the state of the geometrical TLS, the actual figure of merit is the ratio

between the two possible MR values, i.e. ~ 2 . While this is not a bad value, note that in an extreme case one could have record-but-essentially-useless MR values for both geometrical states if they were identical (thus useless as a witness for the geometrical state) and with a fast enough spin dynamics that impedes their spin bistability.

1d) Very often, Dy^{3+} displays a $\pm 15/2$ ground doublet with typical g values close to (20,0,0). If that was the case, the total magnetic moment would be insensitive to the geometric TLS, since very different coordination environments can produce the same spin doublet. The fact that here the authors measure g values around (8,4,2), meaning intermediate and/or mixed M_j values, confirms that in this case it is possible to connect change in coordination environment with a change in the magnetic moment. Perhaps this point could be made more explicitly.

2: This may be misunderstanding on my part, but if (a) the only (or main) effect of the gate field is to lower the barrier, but (b) both the barrier and the energy difference between the two states are of the order of decades of meV except at a particular external field and (c) the whole experiment is $\sim 2K$... doesn't the switching require some fine-tuning of the applied field or its time-dependence? For the same reasons that the non-volatile claim is valid, when the transition is actually desired, I'd expect a specific action needs to be taken. Probably if rather than the generic Figure 2g we could see the actual behavior of the system, the doubt would be solved.

Reviewer #3 (Remarks to the Author):

The authors reported observation of two magnetic states of a single $Dy@C84$ molecule which can be tuned by the electric field effect. They also observed a large magnetoresistance of $\sim 1000\%$ at the resonant tunneling point in their device. DFT calculations indicated that the stable on-volatile switching of the two states originates from an energy barrier of ~ 150 meV, which can be tuned by the electric field, leading to magnetic state transitions. Therefore, the results are interesting and will shed light on further exploration of information storage by switching magnetic state of single atoms. Before its further consideration, here are some issues need to be addressed:

- 1) What is the typical size of the nanogap? How many $Dy@C84$ molecules will reside in the nanogap and be active in the electrical transport?
- 2) The nanogap are formed after drop diluted $Dy@C84$ solution. If the $Dy@C84$ solution is dropped after the nanogap formation, will the electrical transport properties change? Why?
- 3) Au has strong interaction with $Dy@C84$ molecules. Therefore, when performing DFT calculations, the charge transfer effect between Au electrode and $Dy@C84$ molecules needs to be considered.
- 4) In Fig. 2c and 2f, the energy values on the vertical axes are missing. The panels are thus confusing: the data points in Fig. 2c,f are extracted from the dI/dV maps in Fig. 2b,e, but they seem to vary significantly from the original plot. Please clarify.
- 5) In Fig. 2e, the white line of EN is distant from the peaks of dI/dV . The authors should replot the Zeeman shift of EN and reanalyze the corresponding g^* .
- 6) The data seems to be collected from a single device. Are the results reproducible in other

Dy@C84 molecule devices? For example, are the two magnetic states intrinsic states of the Dy@C84 molecule or dependent on the electrode-molecule coupling.

RESPONSE TO REVIEWERS' COMMENTS(NCOMMS-23-50571)

We sincerely appreciate the thorough evaluation of our work by all Reviewers and Editors and the time they spent on our manuscript. In the revised manuscript and Response Letter, we have essentially addressed all of the questions and comments raised by the Reviewers. We think that the resulting study is now significantly improved. Below, we provide a detailed point-by-point response (marked in blue) to each comment. The significant text changes are highlighted yellow in the revised manuscript.

Reviewer #1 (Remarks to the Author)

Comments:

The paper 'Electrically controlled non-volatile switching of single-atom magnetism in a Dy@C84 single-molecule transistor' address an interesting and forefront subject of research, however it is so baddley written and as consequence it is difficult if not impossible to review.

Reply: We greatly appreciate Reviewer #1 for the comments on the *significance* of our work despite the bad writing quality. The manuscript was meticulously rewritten to incorporate specific revisions addressing the concerns raised by the reviewer and eliminate any ambiguity to substantiate our conclusions. Additionally, we asked the assistance of editors of Nature Research Editing Service to improve the manuscript's quality. We provide the detailed point-by-point response to reviewer's concerns below.

(1) This can be noticed since the incipit. In the abstract, the Authors state 'The atomic manipulation and manufacturing have for some time been transformative, and represented the frontier of research in the nanotechnology of high-density storage.'

For example they do not specify to which storage they refer to, presumable they refer to data storage.

Reply: We are grateful to the reviewer to point out our inappropriate statement. We agree with the reviewer's evaluation that the "storage" is not accurate. What we refer to is data storage or information storage, especially magnetic data storage, that is, using

the magnetic bistability of magnetic atoms to encode data information. We have also carefully checked and modified our statements to ensure professionalism and logicity.

Action: We have revised the statement of data storage in the manuscript.

(2) The experimental conditions under which the data reported in Figure 2 (a and d) are acquired are not clear at all and this renders rather difficult to follow the rest of the manuscript.

Reply: We are very thankful to the reviewer for this comment to help us significantly improve our work. We carefully revised the description of the experiment, adding details and instructions to make it clear and appropriate. We reorganized the logical sequence of the experiments, so that our article is fully coherent in terms of graphical expression.

I suggest a careful rewriting of the manuscript before it can undergo to a review process.

Reply: We sincerely thank the reviewer for his/her encouraging comments on our work, pointing out our shortcomings and providing us with valuable opportunities to further address the quality of the manuscript and the significance of the research. We have thoroughly responded and revised our manuscript, particularly focusing on enhancing the clarity and precision of the experimental process and results description. The manuscript was also edited by Nature Research Editing Service to ensure the appropriate level of formality and professionalism. Consequently, we are confident that we have significantly elevated the quality of our work, thereby generating substantial interest within the wider community.

Below we list the main revisions in the manuscript.

(i) **Renaming the bistable states.**

We named the two bistable states as magnetic states in the original manuscript referring to a study on magnetic signatures of $N@_{60}$ where the magnetic character of

N@C₆₀ has been identified by a spin state transition under the magnetic field [*Nature Materials* **7**, 884-889 (2008)]. Here, we want to rename these two bistable states as two molecular states because their physical picture stems from the two metastable positions of the Dy atom within the carbon cage. The transition of the Dy atom between the two sites results in a change in coordination environment and subsequently the orbital, the magnetic moment, and the metal-cage hybrid state. Hence, the application of an electric field enables the nonvolatile switching of the bistable molecular states and correspondingly the magnetic moment of the Dy atom.

(ii) The modification of Fig. 2 in the manuscript.

In addition to updating the description of experimental conditions in Fig. 2, we also modified the coordinates of the figure. The data processing process has been thoroughly examined to ensure accuracy and the ambiguous parts in Fig. 2(a-e) have been rectified. The data points in Fig. 2 (c, f) are extracted from the peaks of the second derivative of current (d^2I/dV^2), which is obtained by numerically differentiating the second derivative of the I - V curves. The reason for using the d^2I/dV^2 coloured maps is to provide a clearer representation of the evolution of distinct states in response to magnetic fields. Here we plot the d^2I/dV^2 coloured maps (as shown in Fig. R1.1(b, e)) to illustrate the Zeeman shift for each state for clarity, providing an enhanced visualization compared to the dI/dV maps (Fig. R1.1(a, d)). This d^2I/dV^2 maps or current maps as reported in previous studies were also employed to illustrate the evolution of electronic or spin states with magnetic fields [*Nature* **468**, 1084-1087 (2010); *Nano letters* **16**, 7509-7513 (2016); *ACS nano* **11**, 5879-5883 (2017)] as the differential conductance maps do [*Physical review letters* **121**, 037703 (2018)]. Comparing the dI/dV and d^2I/dV^2 maps for state 2, the Zeeman shift of E_N is much more blurred in the differential conductance map, and it is not possible to clearly distinguish whether there is a splitting of the energy level, which is clearly demonstrated in the d^2I/dV^2 map.

Figure R1.1 | Zeeman effect of the Dy@C₈₄ SMT and variation in single-atom magnetism. (a, d) Colored maps of dI/dV as a function of B and V_{sd} for state 1 (a) and state 2 (d) at constant V_g ($V_g = -7.1V$ for state 1 and $-7.7V$ for state 2, and just to the right of the degeneracy point marked by the white dashed lines parallel to the vertical axis in Fig. 2(a, d)). (b, e) The d^2I/dV^2 maps as a function of B and V_{sd} for state 1 (a) and state 2 (d) at the same positions, providing clearer indications of energy shifts in the ground and excited states with B . Note that the excited state of state 2 exhibits a clear split-like behaviour, probably due to the crossover between the $N-1$ and N excited states rather than real energy splitting under magnetic field. (c, f) The relative energies of the ground and excited states for state 1 and state 2 plotted as a function of B , which are extracted from the positions of the d^2I/dV^2 peaks in (b) and (e). We then fit the data linearly according to the Zeeman effect, and the values of g^* are shown in the figure, which also can be expressed as the effective magnetic moments (μ) of the ground and excited states.

To ensure the accuracy of data processing, we extracted the data points from the peaks of the dI/dV and d^2I/dV^2 maps, respectively. Subsequently, curve fitting is

employed to determine the effective g -factors for different states. The data points corresponding to state 1 recorded prior to 1.5 T exhibit significant variability in their behavior; hence, these "outlier" data points are excluded during the fitting process. The g^* values for each state are summarized in Table R1.1, and both fits demonstrate excellent agreement considering the fitting error.

Table R1.1 Comparison of the fitted effective g -factor (g^*) for different states according to the Zeeman effect derived from d^2I/dV^2 peaks and dI/dV peaks.

Molecular State		Effective g -factor (g^*) derived from d^2I/dV^2 peaks	Effective g -factor (g^*) derived from dI/dV peaks
State1	ES(N)	-7.0 ± 0.1	-7.2 ± 0.1
	GS($N-1$)	2.9 ± 0.2	3.3 ± 0.4
	GS(N)	-3.8 ± 0.3	-4.1 ± 0.2
State2	ES(N)	-8.4 ± 0.4	-
	GS($N-1$)	2.2 ± 0.2	2.4 ± 0.2
	GS(N)	-5.1 ± 0.3	-5.5 ± 0.4

(iii) Switching operation

The schematic diagram of bistable state switching has also been added in the manuscript to avoid ambiguity. The bistable states in the experiment are switched by scanning the gate voltage forward to 10 V (or backwards to -10 V) to state 1 (or state 2) while applying a small bias voltage (5 mV) at 1.8 K (Fig. R1.2). The threshold value of the switching gate voltage is ± 10 V, the scan step is ~ 0.1 V/s and the switching time is in the order of seconds. During the process of molecular state switching by an electric field, a transition in the magnetism of the Dy atom also occurs correspondingly.

Figure R1.2 | Switching operation of the two molecular states. (a) The two molecular states are switched by scanning the gate voltage forward to 10 V (or backwards to -10 V) while applying a small bias voltage (5 mV) at 1.8 K. (b) Schematic diagram of switching between the two molecular states. The switching gate voltage is ± 10 V.

(iv) **The reaction path between the two bistable states and the relative angle between their electric dipoles.**

We have made modifications to Fig. 4d in the manuscript to enhance its clarity in illustrating the angle between the electric dipoles of the two bistable states, as well as the angles between the electric dipole moments of Dy@C₈₄ and the direction of the set electric field in our calculations. Figure R1.4 shows the relative angle between the two electric dipoles of 168.4° . The orientation of the electric dipole moment is also crucial, enabling switch and memory operations through reorientation of a single electric dipole. [*Nature Nanotechnology* **15**, 1019-1024 (2020); *Nature materials* **21**, 917-923 (2022)].

Figure R1.3 | The electric dipole moments of state1 ($0.70 e\text{\AA}$) and state2 ($0.64 e\text{\AA}$)

with the relative angle (168.4°) between two electric dipoles. The angles between the electric field and the two dipoles are 17.5° (state 1) and 3.9° (state 2), respectively. The direction of the electric field is set to coincide with the directions of the two electric dipole moments.

Figure R1.4 | Reaction path from state 1 to state 2. (a, b) The energy profile with structural diagrams of the Dy ion's position (a) and the dipole moment in the z-direction (b) when climbing along the reaction path.

Furthermore, Fig. 4b in the manuscript illustrates the energy barrier between two bistable states. To gain further insights into the transition process between the states, we conducted theoretical calculations using CI-NEB (climbing image nudged elastic band) method to determine the geometrical reaction path from state 1 to state 2 [*J. Chem. Phys.* **113**, 9901-9904 (2000); *J. Chem. Phys.* **113**, 9978-9985 (2000)]. The reaction path of the Dy ion and the dipole moment in the z-direction between the two bistable states during climbing along the energy barrier is illustrated in Figure R1.3. The evolution of both the energy profile with molecular structure and dipole moment along

the z-axis (the designated transport direction) provides valuable insights into the trace of the Dy atom. The z-components of the dipole moments at state 1 and state 2 are approximately $-0.6 e\text{\AA}$ and $0.45 e\text{\AA}$, respectively. When a positive electric field is applied, the Dy atom prefers the orientation of state 1, while state 2 under a negative electric field.

(v) Charge transfer and coupling between Au electrodes and molecules.

In the manuscript, we investigated the switching mechanism of two bistable states in free neutral Dy@C₈₄ molecules using theoretical calculations. The resulting calculations reveal that the gate electric field effectively reduces the energy barrier, leading to a molecular state transition, accompanied by a displacement in the position of the Dy atom, rearrangement of molecular orbitals, and charge redistribution. Noteworthy, the energy barrier between the two molecular states in free neutral molecules is approximately 150 meV and the energy difference of the two states is 60 meV. Considering a silicon oxide layer thickness of 10–30 nm and a switch gate voltage of 10 V, the estimated switching gate electric field is around 0.03–0.1 V/Å. Furthermore, with a system temperature of approximately 2 K and a bias voltage of 5 mV, the barrier and the energy difference between the two states seem to be orders of magnitude larger than those observed in experimental measurements. Therefore, it is necessary to consider other factors that impact the energy of the two states and the barrier between them. These factors include variations in molecular charge state, interactions between Dy@C₈₄ and Au electrodes (as indicated in Table R1.2), as well as additional influences from surrounding molecules and electric fields along the current direction.

Table R1.2 Energy differences between the two molecular states and energy barriers of bistable switching under different conditions. Charge transfer and the interaction with electrodes play an important role.

	Conditions	Energy differences (meV)	Barrier (meV)

Charge transfer	Neutral	+61	+145
	Charge = +1	+120	+134
	Charge = +2	+87	+125
	Charge = -1	-1	+246
	Charge = -2	+103	+266
Interaction with Au electrodes	Au ₁₆ electrode	+2	+92

Firstly, the Dy@C₈₄ molecule may have charge transfer with the gold electrodes, which leads to change of the charge state. The calculation results reveal significant disparities in energy and energy barrier between the bistable states for different charge states. As illustrated in Table R1.2, adding an integer number of electrons will significantly increase the energy barrier between the bistable states, while reducing the number of electrons may decrease the barrier. Moreover, the energy difference between two bistable states varies considerably with the charge state. For instance, the energy barrier is reduced to 125 mV when charge = +2.

Secondly, we recalculated the energy barrier when the molecule coupled to the Au electrodes, because there is Au-C covalent bonding [*Nature Reviews Materials* **1**, 1-15 (2016)] between the Dy@C₈₄ molecule and Au electrodes, and the strong interaction will have an important impact on molecular energy, energy barrier and electrical transport properties. Utilizing a Au₁₆ electrode model, our calculations demonstrate a significant reduction in the energy difference between the two bistable states to 2 meV, as well as a decrease in the energy barrier to 92 meV, originated from the coupling between the molecule and Au₁₆ cluster (see Fig. R1.5). Besides, the applied electric field along the current direction and the interaction with a neighboring molecule [*Nature materials* **21**, 917-923 (2022)] will also slightly reduce the energy barrier. Our calculations show that applying an electric field of 0.05 V/Å parallel to the transport direction can reduce the energy barrier by 5 meV.

The situation in the experiment may be more complicated. The electrode model of Au₁₆ may not accurately describe the actual electrode morphology and molecule-

electrode coupling, accompanied with an uncertain charge transfer, resulting in a partial inconsistency between experimental results and theoretical calculations, which has also been reported in other work [*Nature materials* **21**, 917-923 (2022)]. However, considering the above factors, it can be inferred that the switching gate electric field can reduce the energy barrier between the bistable states to the order of meV.

Figure R1.5 | Energy barriers under different gate electric fields in Dy@C84 molecules free of electrodes and coupled to Au electrodes. (a, b) Calculated energy barrier under different gate electric fields between the bistable states free of electrodes (a) and coupled to Au electrodes (b). The interaction between the molecule and electrodes leads to significant reductions in the energy difference and the barrier. The gate electric field can effectively lower the energy barrier, thereby enhancing the transition probability between the two molecular states.

Reviewer #2 (Remarks to the Author)

Comments:

The result is highly interesting and in my opinion deserves publication in Nature Communications. That being said, let me offer a couple of comments and questions:

Reply: We highly appreciate the reviewer for evaluations on the novelty and significance of our work. With the supplementary calculations, discussion, and clarification of physical picture, our work, we think, is much improved. The following is our point-by-point response and the detailed discussion to address the concerns raised.

(1) I think primarily referring to the two states in the system as "two magnetic states" is dangerously misleading and should be avoided. While indeed the two states present a different magnetic moment, and while the two states present different values of magnetoresistance, the nature of the two-level-system (TLS) is fundamentally not magnetic, rather geometrical: the Dy is in either of two possible positions in space. Naturally, this affects the electric dipole both in direction and in magnitude, and also the coordination around the Dy, so in principle also the magnetic dipole can be expected to be different.

Reply: We are very grateful to the reviewer for this important and meaningful comment. As mentioned in the study of an endofullerene N@C₆₀ device, the magnetic character of N@C₆₀ has been identified by a spin state transition under the magnetic field [*Nature Materials* **7**, 884-889 (2008)]. Therefore, we called these two bistable states as magnetic states in the original manuscript. Now, we have seriously considered and totally agree with the viewpoint of the reviewer that "the magnetic state" is dangerously misleading. To avoid ambiguity and misleading, we rename these two bistable states as two molecular states. Their physical picture stems from the two metastable positions of the Dy atom within the carbon cage, which is fundamentally geometrical. The orbital of Dy³⁺ in Dy@C₈₄ is confined within the carbon cage and interacts with it, the transition of the Dy ion between the two sites results in a change in coordination environment and subsequently altering the orbital, the magnetic moment, and the metal-cage hybrid state.

Although the magnetic bistability of individual Ho atoms on MgO substrates and

individual Dy atoms on MgO/Ag(100) were observed [*Science* **352**, 318, (2016); *Nature* **543**, 226-228 (2017); *Nat. Commun.* **12**, 4179, (2021); *ACS Nano* **16**, 11182–11193, (2022)]. However, achieving the magnetic bistability of Dy in the carbon cage is challenging due to the requirement for a combination of strong spin-orbit coupling and an appropriate crystal field. Dy@C₈₂ has shown paramagnetic properties even at 1.8 K [*The Journal of Physical Chemistry B* **104**, 1473-1482 (2000)]. Therefore, magnetic bistable states like in a single-ion magnet, which is protected and stabilized by the magnetic anisotropy, is unlikely to exist in Dy@C₈₄. The additional calculation results reveal an energy difference of ~ 500 μeV between magnetic moments oriented along two orthogonal directions, indicating that Dy@C₈₄ is more likely to exhibit paramagnetic behavior.

The bistable molecular states arise from the transition of the Dy ion between the two sites, accompanied by a change in coordination environment, rearrangement of molecular orbitals, redistribution of charge, and alteration of magnetic moment.

Elaborating further:

(1a) DFT calculations could show the geometrical "reaction path" with the lowest activation energy between "reactives" (state 1) and "products" (state 2): how does the Dy ion move? This would basically translate into improving the information provided in the current version of Fig 4b.

Reply: We thank the reviewer for this important question on reaction path between the two states and the kind suggestion. We carried out theoretical calculations on the geometrical reaction path between state 1 and state 2 using the CI-NEB (climbing image nudged elastic band) calculations [*J. Chem. Phys.* **113**, 9901-9904 (2000); *J. Chem. Phys.* **113**, 9978-9985 (2000)]. Figure R2.1 shows the reaction path of movement of the Dy ion and the dipole moment in the z-direction between the two bistable states when climbing along the energy barrier. The evolution of the energy profile and the dipole moment along the z-axis (the set transport direction) provides more information about the Dy atom's trace within the carbon cage. The z-components of the dipole moments at state 1 and state 2 are approximately $-0.6 e\text{\AA}$ and $0.45 e\text{\AA}$, respectively. When a

positive electric field is applied, the Dy atom prefers the orientation of state 1, while state 2 under a negative electric field.

Figure R2.1 | Reaction path from state 1 to state 2. (a, b) The energy profile with structural diagrams of the Dy ion's position (a) and the dipole moment in the z-direction (b) when climbing along the reaction path.

Action: We added Fig. R2.1 in the manuscript as new Fig. 4(e).

(1b) Movement of a charge between two positions would primarily change the direction of the electric dipole and only secondarily its magnitude. This is represented on Figure 4f, but we could benefit from a focus on the direction, e.g. the relative angle between the two dipoles, or the two relative angles between each of the electric dipoles and the external electrical field, if it can be estimated.

Reply: We thank the reviewer for this important question. We have made modifications

to Fig. 4d in the manuscript to enhance its clarity in illustrating the angle between the electric dipoles of the bistable states, as well as the angles between the electric dipole moments of Dy@C₈₄ and the direction of the set electric field in calculation. As shown in Fig. R2.2, the relative angle between the two electric dipoles is 168.4°. The orientation of the electric dipole moment is also crucial, and switch and memory operations can be achieved by utilizing the reorientation of a single electric dipole. [*Nature Nanotechnology* **15**, 1019-1024 (2020); *Nature materials* **21**, 917-923 (2022)].

Figure R2.2 | The electric dipole moments of state1 (0.70 eÅ) and state2 (0.64 eÅ) with the relative angle (168.4°) between two electric dipoles. The angles between the electric field and the two dipoles are 17.5° (state 1) and 3.9° (state 2), respectively. The direction of the electric field is set to coincide with the directions of the two electric dipole moments.

Action: We modified Fig. 4(d) with a clear illustration of the relative angle (168.4°) between two electric dipoles in the two bistable states.

(1c) In each of the two geometrical states there is a very strong MR, in the order of 10^3 . This may be interesting in itself... if the spin dynamics between each of the two magnetic states within each of the two geometrical states was slow enough to employ it as an additional memory bit. However, if one is just employing the geometrical TLS for memory and the MR as a way of measuring the state of the geometrical TLS, the actual figure of merit is the ratio between the two possible MR values, i.e. ~ 2 . While this is not a bad value, note that in an extreme case one could have record-but-essentially-useless MR values for both geometrical states if they were identical (thus

useless as a witness for the geometrical state) and with a fast enough spin dynamics that impedes their spin bistability.

Reply: We thank the reviewer for raising this very important question and nice suggestion. We compared the very strong MR in the order of 10^3 and its underlying mechanism with other molecular systems. The MR observed in some organic radicals typically arises from strong spin-orbit coupling or interface coupling [*Chemical Society Reviews* **42**, 5907-5943 (2013); *Nano Letters* **22**, 5773-5779 (2022); *Nano Letters* **16**, 4960-4967 (2016); *ACS Nano* **10**, 8571-8577 (2016)]. For instance, the large MR observed in OPT self-assembled monolayers (SAMs) is caused by spin coupling with spin-dependent barriers at the Au/S interface [*ACS Nano* **10**, 8571-8577 (2016)], whereas in the TEMPO-OPE single-molecule device, it arises from a reduction of the interface coupling and loss of electron coherence during tunnelling and reflection [*Nano Letters* **16**, 4960-4967 (2016)] and the MR at PTM junctions is also induced by spin-dependent scattering occurring at the interface [*Nano Letters* **22**, 5773-5779 (2022)]. Additionally, in a Fe₄ single-molecule magnet [*Physical Review B* **91**, 035442 (2015)], the MR extracted from the resonant peak is attributed to the hindrance of high-energy transitions by an external magnetic field, while low-energy transitions remain unaffected to contribute to electron transport within a certain temperature window, but it exhibits a tendency to saturate at high fields within the range of 1-10%. We exclude the interface mechanism and the effect of spin polarization reported in previous negative MR studies [*Nano Letters* **11**, 2634-2639 (2011); *Nature Materials* **10**, 502-506 (2011); *ACS Nano* **9**, 4458-4464 (2015)]. Therefore, we adopt the metal-cage hybrid state mechanism to explain the observed MR behaviours in the Dy@C₈₄ SMT. The magnetic field enhances the overlap gap between the Dy and cage components of the metal-cage hybrid state, thereby significantly reducing the DOS of the hybrid state and consequently suppressing the resonant tunneling current.

The MR behaviour of the bistable states varies due to the sensitivity of the metal-cage hybrid state towards its coordination environment. The ratio of MR can be used as an effective figure of merit to figure out the two geometrical states (the bistable states) under two conditions: firstly, the metal-cage hybrid state is accessible; and secondly,

there exists a substantial disparity between the two hybrid orbitals for the two bistable states. And the ratio is ~ 2 (180%) in our device as described in the manuscript. In fact, the magnetic bistability of individual Ho atoms on MgO was observed, where the two-state signal on Ho atoms shows discrete changes in conductance of typically 2% – 4% using a spin-polarized STM [*Nature* **543**, 226-228 (2017)]. The anisotropic magnetoresistance effect due to magnetic reorientation from in-of-plane to out-of-plane may be more pronounced. For example, FePc molecules deposited on a Au(100) surface exhibit tunneling anisotropy resistance (TAMR) of about -93% at 11 T [*Nature Communications* **10**, 3599 (2019)].

However, achieving the magnetic bistability of Dy in the carbon cage is challenging due to the strict requirement for a combination of strong spin-orbit coupling and an appropriate crystal field. Dy@C₈₂ has shown paramagnetic properties even at 1.8 K [*The Journal of Physical Chemistry B* **104**, 1473-1482 (2000)]. Therefore, magnetic bistable states akin to those found in single-ion magnets, which are protected and stabilized by the magnetic anisotropy, is unlikely to exist in Dy@C₈₄. Additional calculations results reveal an energy difference of ~ 500 μeV between magnetic moments oriented along two orthogonal directions, indicating that Dy@C₈₄ is more likely to exhibit paramagnetic behavior. In this case, the survival of the magnetic state protected by uniaxial magnetic anisotropy becomes challenging, as does the manifestation of anisotropic magnetoresistance behavior upon reorientation of the magnetic moment.

To summarize, Dy@C₈₄ is more likely to exhibit paramagnetic behavior, while the bistability of the magnetic states determined by the magnetic anisotropy of the Dy atom is difficult. However, both experimental and theoretical evidence indicates that the magnetism of the Dy atom changes with the transition between the two molecular states. The observed variation in MR behavior between the two bistable molecular states in our device is likely due to the sensitivity of the metal-cage hybrid state to the coordination environment. The MR ratio can serve as an effective figure of merit for distinguishing the molecular state (the geometric state) if we choose and access a suitable metal-cage hybrid state, which shows a significant MR and a distinguished

variation between the two molecular states. Therefore, the switching of molecular states and single-atom magnetism can be effectively controlled by an electric field, which manifests alterations in the effective magnetic moment and MR behavior.

(1d) Very often, Dy³⁺ displays a $\pm 15/2$ ground doublet with typical g values close to (20,0,0). If that was the case, the total magnetic moment would be insensitive to the geometric TLS, since very different coordination environments can produce the same spin doublet. The fact that here the authors measure g values around (8,4,2), meaning intermediate and/or mixed M_j values, confirms that in this case it is possible to connect change in coordination environment with a change in the magnetic moment. Perhaps this point could be made more explicitly.

Reply: We thank the reviewer for raising this important question and nice suggestion. We agree with the reviewer's viewpoint on the mixed m_j values and that the magnetic moment changes with the coordination field environment. In lanthanide endohedral metallofullerenes (EMF), the orbital of the metal atom could interact with the carbon cage, leading to modifications to the electronic structure and magnetic moment or orbital hybridization [*Chemical Physics Letters* **332**, 219-224 (2000); *Physical review letters* **91**, 185504 (2003); *Physical Review B* **69**, 184421 (2004); *Langmuir* **31**, 11438-11442 (2015)]. The modification of the orbitals of Dy³⁺ or orbital hybridization can be verified by the measurement of the effective g-factor of the ground and excited states for the bistable states in our experiment. The effective g-factors of the ground and excited states are found to be approximately in the range of 2–8, suggests that the ground states are unlikely to be the $|\pm 15/2\rangle$ ground doublet of the Dy³⁺ ion. Instead, they are more likely attributed to mixed m_j states of the Dy 4f orbitals or more likely metal-cage hybrid states. This can be further verified by the fact that there exist disparities in g^* (and thus the effective magnetic moments) between the two bistable states, which is due to transitions of the Dy atom between two different sites inside the carbon cage and can be manipulated by reversible electrical control. The mixed m_j states or the metal-cage hybrid states is more sensitive to the change in coordination environment than the $|\pm 15/2\rangle$ ground doublet of Dy³⁺. It will further cause changes in

the electronic structure, orbital magnetic moment, and ultimately the magnetic moment. Of course, based on the MR behavior of the device, these accessed states are ultimately attributed to metal-cage hybrid states.

The influence of the carbon cage's coordination field on the molecular states can also be supported by theoretical calculations. As indicated in Table R2.1, calculation results demonstrate distinct orbital compositions for the HOMO and LUMO of two molecular states, with the Dy's orbitals contributing 94.9% (state 1) and 94.4% (state 2) to the HOMO, respectively. The above analysis is also elaborated more explicitly in the manuscript.

Table R2.1 Composition in terms of Dy and cage orbitals for the HOMO and LUMO of the bistable states from theoretical calculation.

	State 1	State 2
HOMO	5.1% cage + 94.9% Dy	5.6% cage + 94.4% Dy
LUMO	36.1% cage + 63.9% Dy	37.3% cage + 62.7% Dy

Action: We added Table R2.1 in Supplementary Information as new Table S1.

(2) This may be misunderstanding on my part, but if (a) the only (or main) effect of the gate field is to lower the barrier, but (b) both the barrier and the energy difference between the two states are of the order of decades of meV except at a particular external field and (c) the whole experiment is ~2K... doesn't the switching require some fine-tuning of the applied field or its time-dependence? For the same reasons that the non-volatile claim is valid, when the transition is actually desired, I'd expect a specific action needs to be taken. Probably if rather than the generic Figure 2g we could see the actual behavior of the system, the doubt would be solved.

Reply: We are very thankful to the reviewer for raising this insightful question on the switching operation and energy barrier between the two molecular states. The bistable states in the experiment are switched by scanning the gate voltage forward to 10 V (or

backwards to -10 V) while applying a small bias voltage (5 mV) at 1.8 K to state 1 (or state 2) (see Fig. R2.3). The switching gate voltage is ± 10 V, the scan step is ~ 0.1 V/s and the switching time is in the order of seconds.

Figure R2.3 | Switching operation of the two molecular states. (a) The bistable states are switched by scanning the gate voltage forward to 10 V (or backwards to -10 V) while applying a small bias voltage (5 mV) at 1.8 K. (b) Schematic diagram of switching between two molecular states. The switching gate voltage is ± 10 V.

According to theoretical calculations, the gate electric field effectively reduces the energy barrier, leading to a molecular state transition, accompanied by a displacement in the position of the Dy atom, rearrangement of molecular orbitals, and charge redistribution. Noteworthy, our calculations are based on free neutral molecules without interactions with electrodes, and the energy barrier between the two molecular states is ~ 150 meV. Considering a silicon oxide layer thickness of 10 – 30 nm and a switch gate voltage of 10 V, the switching gate electric field is estimated to be $\sim 0.03 - 0.1$ V/ \AA . In addition, the system temperature is ~ 2 K and the bias voltage is 2 mV. The barrier and the energy difference between the two states are of the order of decades of meV, which are larger than those in experimental measurements.

Therefore, other factors that affect the energy of the two states and the barrier between them need to be taken into consideration. For example, the changes in the molecular charge state, the interaction between the Dy@C₈₄ molecule and gold (Au)

electrodes, as well as other factors such as surrounding molecules and electric fields along the current direction and so on (as shown in Table R2.2).

Firstly, the Dy@C₈₄ molecule may have charge transfer with the electrodes, which leads to change of the charge state. The calculation results reveal significant disparities in energy and energy barrier between the bistable states for different charge states. As illustrated in Table R2.2, adding an integer number of electrons will significantly increase the energy barrier between the bistable states, while reducing the number of electrons may decrease the barrier. For instance, the energy barrier is reduced to 125 mV when charge = +2. Moreover, the energy difference between two bistable states varies considerably with the charge state.

Secondly, there exists a covalent bonding between the Dy@C₈₄ molecule and the Au electrodes, known as Au-C covalent bonding [*Nature Reviews Materials* **1**, 1-15 (2016)]. This strong interaction plays a crucial role in influencing molecular energy, energy barrier, and electrical transport properties. By employing an Au₁₆ electrode model in our calculations, we demonstrate a significant reduction in the energy difference between the two bistable states to 2 meV, as well as a decrease in the energy barrier to 92 meV (see Fig. R2.4).

Besides, the applied electric field along the current direction and the interaction with a neighboring molecule [*Nature materials* **21**, 917-923 (2022)] will also slightly reduce the energy barrier. Our calculations show that applying an electric field of 0.05 V/Å parallel to the transport direction can reduce the energy barrier by 5 meV.

The situation in the experiment may be more complicated. The electrode model of Au₁₆ may not accurately describe the actual electrode shape and the molecule-electrode coupling, and the charge transfer cannot be completely determined, resulting in a partial inconsistency between experimental results and theoretical calculations, which has also been reported in other work [*Nature materials* **21**, 917-923 (2022)]. However, considering the above factors, it can be deduced that the switching gate electric field can reduce the energy barrier between the bistable states to the order of meV.

Figure R2.4 | Energy barriers under different gate electric fields in Dy@C₈₄. (a, b) Calculated energy barrier under different gate electric fields between the bistable states free of electrodes (a) and coupled to Au electrodes (b). The interaction between the molecule and electrodes leads to significant reductions in the energy difference and the barrier. The gate electric field can effectively lower the energy barrier, thereby enhancing the transition probability between the two molecular states.

Table R2.2 Energy differences between the two molecular states and energy barriers of bistable switching under different conditions. Charge transfer and the interaction with electrodes play an important role.

	Conditions	Energy differences (meV)	Barrier (meV)
Charge transfer	Neutral	+61	+145
	Charge = +1	+120	+134
	Charge = +2	+87	+125
	Charge = -1	-1	+246
	Charge = -2	+103	+266
Interaction with Au electrodes	Au ₁₆ electrode	+2	+92

Action: We added Fig. R2.4(b) in the manuscript as new Fig. 4(b) and Table R2.2 in Supplementary Information as new Table S2.

Reviewer #3 (Remarks to the Author)

Comments:

The authors reported observation of two magnetic states of a single Dy@C84 molecule which can be tuned by the electric field effect. They also observed a large magnetoresistance of ~1000% at the resonant tunneling point in their device. DFT calculations indicated that the stable on-volatile switching of the two states originates from an energy barrier of ~150 meV, which can be tuned by the electric field, leading to magnetic state transitions. Therefore, the results are interesting and will shed light on further exploration of information storage by switching magnetic state of single atoms. Before its further consideration, here are some issues need to be addressed:

Reply: We are very grateful to Reviewer #3 for his/her positive comments on our work. He/she thinks that our work is interesting and can be further considered, which we are highly encouraged. We address the reviewer's insightful concerns on experimental details, data dealing and theoretical calculations. We also believe this will be a novel and versatile approach to achieve electrically controlled single-atom magnetic properties in single-molecule transistors and offers promising prospects for integration into high-density data storage devices. Our detailed point-by-point response to the reviewer's comments is shown as follows.

1) What is the typical size of the nanogap? How many Dy@C84 molecules will reside in the nanogap and be active in the electrical transport?

Reply: We thank the reviewer for raising these very important questions. We create nanogaps in spindle-shaped gold nanowires using feedback controlled electromigration break junction (FCEBJ) technique at about 2 K. We can control the conductance of the final break junction to be about 0.02 G_0 , which is related to the nanogap width. The $I(V)$ characteristic curves of the break junctions are measured after the FCEBJ process, and a non-linear curve represents a typical manifestation of tunneling behaviours. The widths of most successfully fabricated nanogaps are estimated to be in the range of 1-2 nm [*Nature* **417**, 722-725 (2002); *JACS* **128**, 2135-2141 (2006); *Nano letters* **11**, 4607-4611 (2011)], which can be deduced by fitting the tunneling current with the Simmons

model [*J. Appl. Phys.* **34**, 1793–1803 (1963); *JACS* **128**, 2135-2141 (2006); *Nano letters* **11**, 4607-4611 (2011)], the relationship between current density J and voltage V is given by:

Where e is electron charge, m is electron mass, h is Planck constant, d and ϕ_0 are the width of the nanogap and the average height of the tunnelling barrier for electrons, respectively. The data of the devices were fitted to the model, revealing nanogaps with widths of approximately 1.2 nm (as depicted in Fig. R3.1), which corresponded to the molecular size of Dy@C₈₄.

The presence of one molecule in the gap is determined by observing gate-dependent Coulomb blockade transport characteristics [*Nature* **417**, 722–725 (2002); *Nano Lett.* **4**, 79–83 (2004); *Nature Mater.* **7**, 179–186 (2008)]. By focusing on a specific resonance tunneling point in the low bias region, one can ensure that only a single molecule is actively involved in the electrical transport [*Nature* **417**, 722–725 (2002); *Nature Mater.* **7**, 179–186 (2008)]. To confirm the presence of only one active molecule bridging the electrodes, after measuring the nonlinear $I(V)$ characteristic curve, we further characterize the junction by examining current at a constant source-drain voltage (e.g., 2 mV) as a function of gate voltage V_g . Subsequently, a stability diagram of differential conductance (dI/dV) is recorded against the source-drain bias V_{sd} and the gate voltage V_g . The presence of one or more Coulomb oscillation peaks in the $I(V_g)$ trace suggests a molecular signal. It is essential to exclude the scenario where the $I(V_g)$ trace exhibits nearly uniform peak characteristics in terms of spacing, height, and FWHM for multiple Coulomb peaks, accompanied by small constant addition energies and nearly homogeneous Coulomb diamonds in the stability map. This phenomenon can be attributed to the signal from a gold cluster contacting with the electrodes and the gate dielectric. And signal of more than one molecule will usually exhibit a blurring or frequent data jump in the gate-dependent Coulomb oscillation peaks.

Therefore, we can distinguish a molecular signal based on the following aspects:

(i) focusing on one clear Coulomb oscillation peak and Coulomb diamond, (ii) the width of the nanogap being in accordance with molecular size, (iii) inherent characteristics of molecules such as vibrations, magnetic excitations, magnetic properties.

Figure R3.1 | Typical size of the nanogap. (a, b) original $I(V)$ data of device A (a) (in the manuscript) and device B (b) (supplementary device) and their fits to the Simmons model. The fitted gap widths are 1.3 nm and 1.2 nm and the tunneling barriers are ~ 2.2 eV and 2.4 eV, respectively. The widths correspond to the molecular size of Dy@C₈₄.

2) The nanogap are formed after drop diluted Dy@C₈₄ solution. If the Dy@C₈₄ solution is dropped after the nanogap formation, will the electrical transport properties change? Why?

Reply: We appreciate the reviewer's important question. The electromigration break junction technique is a main method to fabricate three-terminal single-molecule field effect transistors (FETs), and two routes have been developed including feedback control [*Appl. Phys. Lett.* **86**, 043109 (2005)] and self-breaking schemes [*Applied Physics Letters* **90**(13) (2007)]. The feedback controlled electromigration break junction technique is widely used to investigate single-molecule junctions (SMJs) [*Nature Reviews Physics* **1**, 381-396 (2019); *Physical review letters* **96**, 017205 (2006); *Science* **315**, 77-81 (2007); *Nature* **453**, 633-637 (2008); *Nature* **462**, 1039-1043 (2009); *Nature chemistry* **1**, 635-641 (2009); *Physical Review Letters* **115**, 138302 (2015);

Nature Photonics **12**, 608-612 (2018); *Nature Communications* **14**, 7486 (2023)], which we also adopted in device fabrication as described in the Methods section of the manuscript. Specifically, diluted Dy@C₈₄ solution was deposited onto pre-cleaned Au nanowires, followed by the FCEBJ process at a cryogenic temperature. The size of the final break junction can be effectively controlled by setting and adjusting its final conductance, increasing the probability of forming SMJs. Noteworthy, Au as the preferred material for the nanoelectrode is due to its noble properties, excellent electrical conductivity, and good connectivity with many anchor groups both mechanically and electronically including Au-C covalent bonding [*Nature Reviews Materials* **1**, 1-15 (2016)]. However, the nanoelectrode is rendered unstable at room temperature due to the high atomic mobility of Au [*Appl. Phys. Lett.* **94**, 123108 (2009)]. If the break junction process is carried out at room temperature to achieve a desired nanometer-scale gap, thermal perturbation significantly enhances the migration rate of Au atoms on the surface of SiO₂, resulting in an enlargement of the nanoscale gap. In this case, it becomes challenging to form a stable SMJ if breaking the junction first and subsequently dropping the molecular solution. Therefore, dropping the molecular solution in advance has become a common approach.

For the self-breaking scheme, it takes advantage of the high mobility of gold atoms to self-break the junction at room temperature [*Applied Physics Letters* **90**(13) (2007)]. There are also many experiments using this method [*Nano letters* **10**, 105-110 (2010); *Physical review letters* **109**, 147203 (2012); *Nano letters* **15**, 3109-3114 (2015); *Physical Review Letters* **118**, 117001 (2017); *Nature Nanotechnology* **16**, 426-430 (2021)]. The gold nanowires are initially narrowed down to a few conductance quanta using an actively controlled electromigration technique, and subsequently undergo self-breaking for hours at room temperature to form nanoscale gaps after removal of the applied voltage. In this way, it is workable with or without changing the order of dropping molecular solutions. The formation of single molecular junctions can potentially occur by depositing molecular solutions onto the fabricated nanogaps or immersing them in molecular solutions for a period of time.

In fact, these two routes do not alter the nature of Au-C covalent bonding. We have

explored both routes but may not have a good grasp of the parameters and skills of the self-breaking, with the former approach having a little high formation possibility of SMTs through controlling the final conductance and excluding the signal of gold clusters. The nature of Au-C covalent bonding is expected to remain unaffected by the order in which the molecular solution is dropped. However, factors such as the shape of the electrodes, anchoring site, and local electrostatic environment of the molecule may exert a significant influence on electrical transport [*Science* **292**, 2303-2307, (2001); *Nature* **442**, 904-907, (2006); *Nature Nanotechnology* **2**, 176-179, (2007); *Physical Review Letters* **111**, 246806, (2013); *Nature nanotechnology* **8**, 282-287, (2013)]. These factors differ a lot with devices.

3) Au has strong interaction with Dy@C₈₄ molecules. Therefore, when performing DFT calculations, the charge transfer effect between Au electrode and Dy@C₈₄ molecules needs to be considered.

Reply: We are very thankful to the reviewer for raising this insightful question on the interactions and the charge transfer effect between Au electrode and molecules. According to our theoretical calculations, the gate electric field effectively reduces the energy barrier, leading to a molecular state transition, accompanied by a displacement in the position of the Dy atom, rearrangement of molecular orbitals, and charge redistribution. Furthermore, in response to the reviewer's suggestion, we have taken the influence of the interaction between Au electrode and Dy@C₈₄ molecules into account by considering charge transfer and electrode-molecule coupling separately. This approach allows us to explore the impact of the electrode on the bistability of the molecular states. Theoretical calculations reveal a significant effect of these factors on both energy difference and energy barrier between the two bistable states (as shown in Table R3.1).

Charge transfer: The charge transfer between the Dy@C₈₄ molecule and the gold electrodes leads to the change of the charge state of the molecule. Although the precise quantification of the transferred charge is challenging. The calculation results reveal significant disparities in energy difference and energy barrier between the bistable states

for varying charge states. As illustrated in Table R3.1, the addition of an integer number of electrons leads to a substantial increase in the energy barrier, whereas a reduction in electron number may result in barrier reduction. For instance, the energy barrier is reduced from 145 meV for neutral free molecules to 125 mV when charge = +2. Moreover, the energy difference between two bistable states varies considerably with the charge state.

Electrode-molecule coupling: There exists Au-C covalent bonding [*Nature Reviews Materials* **1**, 1-15 (2016)] between the Dy@C₈₄ molecule and Au electrodes, and this strong interaction will have an important impact on molecular energy, energy barrier and electrical transport properties. Utilizing a Au₁₆ electrode model, our calculations demonstrate a significant reduction in the energy difference between the two bistable states to 2 meV, as well as a decrease in the energy barrier to 92 meV, originated from the coupling between the molecule and Au₁₆ cluster (see Fig. R2.4). The situation in the experiment may be more complicated. The electrode model of Au₁₆ may not accurately describe the actual electrode shape and the molecule-electrode coupling, and the charge transfer cannot be completely determined. Consequently, a partial variance between experimental findings and theoretical computations arises, as has been previously documented in other studies [*Nature materials* **21**, 917-923 (2022)].

Figure R3.2 | Energy barriers under different gate electric fields in Dy@C₈₄

molecules. (a, b) Calculated energy barriers under different gate electric fields between the bistable states free of electrodes **(a)** and coupled to Au electrodes **(b)**. The interaction between the molecule and electrodes leads to significant reductions in the energy difference and the barrier. The gate electric field can effectively lower the energy barrier, thereby enhancing the transition probability between the two molecular states.

Therefore, considering the charge transfer and electrode-molecule coupling, both factors exert a significant influence on the energy difference and barrier of the bistable states. Although there is not complete quantitative consistency between theory and experiment, it can be inferred that the switching gate electric field can effectively reduce the energy barrier between bistable states to an order of meV.

Table R3.1 Energy differences between the two molecular states and energy barriers of bistable switching under different conditions. Charge transfer and the interaction with electrodes play an important role.

	Conditions	Energy differences (meV)	Barrier (meV)
Charge transfer	Neutral	+61	+145
	Charge = +1	+120	+134
	Charge = +2	+87	+125
	Charge = -1	-1	+246
	Charge = -2	+103	+266
Interaction with Au electrodes	Au ₁₆ electrode	+2	+92

4) In Fig. 2c and 2f, the energy values on the vertical axes are missing. The panels are thus confusing: the data points in Fig. 2c,f are extracted from the dI/dV maps in Fig. 2b,e, but they seem to vary significantly from the original plot. Please clarify.

Reply: We are grateful to the reviewer to point out the inaccuracy of vertical axes as well as our inappropriate data dealing. The data points in Fig. 2(c, f) are actually extracted from the peaks of the second derivative of current (d^2I/dV^2), which is obtained

by numerically differentiating the second derivative of the I - V curves. We have conducted careful data verification to ensure accuracy. Here we plot the d^2I/dV^2 (as shown in Fig. R3.2(b, e)) to illustrate the Zeeman shift for each state for clarity, providing an enhanced visualization compared to the dI/dV maps (Fig. R3.2(a, d)). Consistent with previous studies, in addition to differential conductance maps or curves describing the evolution of electronic or spin states with magnetic fields [*Physical review letters* **121**, 037703 (2018)], current or d^2I/dV^2 maps were also used to show the response with magnetic field [*Nature* **468**, 1084-1087 (2010); *Nano letters* **16**, 7509-7513 (2016); *ACS nano* **11**, 5879-5883 (2017)]. When comparing the dI/dV and d^2I/dV^2 maps for state 2, it is observed that the Zeeman shift of E_N appears more diffuse in the dI/dV map, making it challenging to clearly discern any splitting of the energy level. Conversely, this distinction is evident in the d^2I/dV^2 map.

To ensure the accuracy of data processing, the data points are extracted from the peaks of the dI/dV and d^2I/dV^2 maps, respectively. Subsequently, curve fitting is employed to determine the effective g -factors for different states. The data points of state 1 recorded before the 1.5 T exhibit significant variability in their behavior. Therefore, these “bad” data points are excluded during the fitting process. The g^* values for each state are summarized in Table R3.2, and both fits demonstrate excellent agreement when considering the fitting error.

Figure R3.3 | Zeeman effect of the Dy@C₈₄ SMT and variation in single-atom magnetism. (a, d) Coloured maps of dI/dV as a function of B and V_{sd} for state 1 (a) and state 2 (d) at constant V_g ($V_g = -7.1$ V for state 1 and -7.7 V for state 2, on the right of the degeneracy point marked by the white dashed lines parallel to the vertical axis in Fig. 2(a, d)). (b, e) The d^2I/dV^2 maps as a function of B and V_{sd} for state 1 (a) and state 2 (d) at the same positions, providing clearer indications of energy shifts in the ground and excited states with B . Note that the excited state of state 2 exhibits a split-like behaviour, probably due to the crossover between the $N-1$ and N excited states rather than real energy splitting under the magnetic fields. (c, f) The relative energies of the ground and excited states for state 1 and state 2 plotted as a function of B , which are extracted from the positions of the d^2I/dV^2 peaks in (b) and (e). We then fit the data linearly according to the Zeeman effect, and the values of g^* are shown in the figure, which also can be expressed as the effective magnetic moments (μ) of the ground and excited states.

Table R3.2 Comparison of the fitted effective g -factor (g^*) for different states according to the Zeeman effect derived from d^2I/dV^2 peaks and dI/dV peaks.

Molecular State		Effective g -factor (g^*) derived from d^2I/dV^2 peaks	Effective g -factor (g^*) derived from dI/dV peaks
State1	ES(N)	-7.0 ± 0.1	-7.2 ± 0.1
	GS($N-1$)	2.9 ± 0.2	3.3 ± 0.4
	GS(N)	-3.8 ± 0.3	-4.1 ± 0.2
State2	ES(N)	-8.4 ± 0.4	-
	GS($N-1$)	2.2 ± 0.2	2.4 ± 0.2
	GS(N)	-5.1 ± 0.3	-5.5 ± 0.4

Action: We added Fig. R3.2(b, c, e, f) as new Fig. 2(b, c, e, f) in the manuscript and modified Table 1 with Table R3.2.

5) In Fig. 2e, the white line of EN is distant from the peaks of dI/dV . The authors should replot the Zeeman shift of EN and reanalyze the corresponding g^* .

Reply: We appreciate the reviewer's suggestion about checking the Zeeman shift of E_N . As we mentioned in **Reply 4**, the Zeeman shift of E_N for state 2 is more obvious in the d^2I/dV^2 map than the dI/dV map. As shown in the differential conductance map (Fig. R4(d)), the broadening of the E_N level with increasing magnetic field cannot be attributed to the Zeeman shift of a single energy level. Instead, based on a clear visualization provided by the d^2I/dV^2 map (Fig. R4(e)), it is likely that this broadening arises from the movement of two energy levels that are initially closely spaced and indistinguishable at zero field but gradually separate under different magnetic responses. We do not attribute them as splitting of the same level due to both levels exhibiting down shifts with respect to the magnetic fields. The split-like behaviour with B is likely a result of the crossover between the $N-1$ and N excited states. The corresponding effective g -factors are 8.4 and 1.2, respectively.

Figure R3.4 | Comparison of the dI/dV map and the d^2I/dV^2 map of State 2 for the Dy@C₈₄ SMT. (a, b) Colored maps of the dI/dV (a) and the d^2I/dV^2 (b) as a function of B and V_{sd} for state 2 at constant V_g ($V_g = -7.7$ V). The d^2I/dV^2 map provides a clearer indication of energy shifts in the ground and excited states with B . Note that the excited state of state 2 exhibits a clear split-like behaviour, probably due to the crossover between the $N-1$ and N excited states rather than real energy splitting under magnetic field.

6) The data seems to be collected from a single device. Are the results reproducible in other Dy@C₈₄ molecule devices? For example, are the two magnetic states intrinsic states of the Dy@C₈₄ molecule or dependent on the electrode-molecule coupling.

Reply: We greatly appreciate the reviewer for raising these important issues regarding the experimental validation whether the results are reproducible in other devices. We want to rename the bistable states as the molecular states. As mentioned in a study of an endofullerene N@C₆₀ device, the magnetic character of N@C₆₀ has been identified by a spin state transition under the magnetic field [*Nature Materials* **7**, 884-889 (2008)]. Therefore, we called these two bistable states as magnetic states in the original manuscript. However, their physical picture stems from the two metastable positions of the Dy atom within the carbon cage. The orbital of Dy³⁺ in Dy@C₈₄ is confined within the carbon cage and interacts with it, the transition of the Dy ion between the two sites

results in a change in coordination environment and subsequently altering the orbital, the magnetic moment, and the metal-cage hybrid state. To avoid ambiguity and misleading, we rename these two bistable states as two molecular states.

Although the coupling conformation and electrostatic environment vary with devices, which leads to the difficulty of completely consistent transport properties of devices [*Science* **292**, 2303-2307, (2001); *Nature* **442**, 904-907, (2006); *Physical Review Letters* **111**, 246806, (2013); *Nature nanotechnology* **8**, 282-287, (2013)], we have successfully reproduced the intrinsic properties of single-atom magnetism in another Dy@C₈₄ SMT. We fabricated another device (device B) and conducted measurements on its electrical transport properties as well as its response to an external magnetic field. The whole transport measurement was carried out at a cryogenic temperature of 1.8 K. Figure R3.5(a) shows the $I(V_{sd})$ characteristic curves at different gate voltages V_g , while in R3.5(b) the curves of the differential conductance (dI/dV) with respect to V_g are plotted at zero bias ($V_{sd} = 0$ mV). They reveal a sequential electron tunnelling through the Dy@C₈₄ molecule, which exhibits Coulomb blockade in the low bias region and two set of Coulomb oscillation patterns (the black line and the red line). By varying V_g , we were able to probe various charge states labelled by different N or molecular orbitals through single-electron tunnelling (SET). Nevertheless, precisely aligning the molecular orbitals with the Fermi surface of the metal electrodes is challenging due to image charges' uncertainty. Figure R3.4(c, d) show two coloured maps of the dI/dV in device B as a function of V_{sd} and V_g corresponding to the black line ($V_g = 2 - 14$ V) and the red line ($V_g = -14 - 2$ V), respectively. Similar to device A, we also attribute these two Coulomb oscillation patterns to two bistable molecular states, which can be derived from the different excited state energies and responses to the magnetic field as shown and discussed below.

These Coulomb stability diagrams of the two states with different diamonds are dominated by SET. The intersections of these Coulomb edges at zero bias are the charge degeneracy points or the resonant tunneling points. We define the state represented by the black line as molecular state 1 (state 1) and the red line as state 2. The gate efficiency factor α is 0.0076 for state 1 and 0.008 for state 2. The differences also manifest in the

locations of degeneracy points and the energies of excited states. The excited states at about 20 meV (state 1) and 27 meV (state 2), marked indicated by red arrows, can presumably be assigned to the longitudinal vibration of the Dy ion coupled to the carbon cage [*Applied Physics Express* **13**, 105002 (2020).], rather than an intrinsic vibrational mode of the carbon cage.

The Zeeman shift of the individual states in device B was confirmed by employing a slightly different measurement method compared to device A. Specifically, we recorded the source-drain current curve (I_{sd}) while sweeping V_g at a fixed bias voltage ($V_{sd} = 2$ mV) under various magnetic fields to simultaneously observe the response of the two resonant tunneling peaks with respect to magnetic field. The evolution of consecutive resonant tunneling peaks with respect to the magnetic field is depicted in Fig. R3.6(a, c), demonstrating clear Zeeman shift of the N charge state and splitting behavior of the $(N+1)$ charge state. We determined the addition energy ($E_{add}(N) = \mu_{N+1} - \mu_N = E_C + E_{N+1} - E_N$) by measuring the spacing ΔV_g between consecutive Coulomb peaks, subsequently converting ΔV_g to energy using the gate efficiency factor α .

Then, we investigated the magnetic field dependence of the addition energy. The charging energy is independent with magnetic field, the dependence of the addition energy on the magnetic field completely depends on that of the level spacing ($E_{N+1} - E_N$). The slope of the addition energy E_{add} with respect to B depends on the relative spin orientation between two neighboring charge states. We define $\Delta\mu_{N+1}$ as the energy level spacing between the $(N+1)$ and N charge states, and Δ_{N+1} as the level splitting of the $(N+1)$ charge state. Figure R3.6(b, d) show the dependence of the addition energy and the level splitting on the magnetic field, and the data are fitted to obtain the effective g -factor. The g^* value for the level splitting Δ_{N+1} of the $(N+1)$ charge state reaches approximately 20, which is comparable to the splitting in the ground state $|\pm 15/2\rangle$ of Dy^{3+} . The ground state of the $(N+1)$ charge state of the molecule is likely to be predominantly contributed by the $|\pm 15/2\rangle$ ground state of Dy^{3+} . The extracted g^* value from the slope of $\Delta\mu_{N+1}$ is approximately -2, indicating that the ground state of the N charge state with magnetic moment of $\sim 8\mu_B$ may originates from either the excited state of Dy^{3+} or more likely a hybridized state with carbon cage.

The Dy@C₈₄ SMT in device B also exhibits magnetoresistance (MR) behaviours near the N resonant tunneling point. In Fig. R3.7(a, c), the evolution of the resonance tunneling peaks for the N charge state are plotted for both states (indicated by black arrows in Fig. R3.5(c, d)) as a function of an external magnetic field. Figure R3.7(b, d) also exhibit quite large positive MR and the amplitude of the peaks are suppressed by the magnetic field. The MR of state 1 can reach 45% at $B = 9$ T, and that of state 2 is 40%. The phenomenon can be effectively explained by our proposed metal-cage hybrid state mechanism, as detailed in our manuscript. As previously mentioned, the magnetic moment of the $(N+1)$ ground state is approximately $10 \mu_B$, which can be attributed to the most contribution of the $|\pm 15/2\rangle$ ground state of Dy³⁺. On the other hand, the N ground state exhibits a magnetic moment that differs from that of the $(N+1)$ state by about $2 \mu_B$, likely originating from the metal-cage hybrid state. The degree of hybridization will affect the MR properties. Therefore, the N charge state exhibits a certain degree of hybridization, which can be figured out from the MR value. However, compared with device A in the manuscript, the MR value is smaller, indicating a lower level of hybridization in this ground state between the higher m_j state of Dy³⁺ and the carbon cage. Additionally, the small variation in the MR properties of the two molecular states confirmed this low degree of hybridization, which exhibits limited sensitivity to the change in the coordination environment.

Consequently, the characteristics of the molecule's ground state exhibit variation across diverse devices due to uncertainty regarding accessible molecular orbitals and thus single atom magnetism. Nonetheless, the intrinsic magnetic behavior still exhibits commonalities. Besides, achieving precise control over single-atom magnetic properties necessitates the utilization of hybridization, which enables more precise control and detection of the single-atom electronic structure, orbital configuration, and the magnetic moment.

Figure R3.5 | Basic electrical transport measurements of device B. (a) The I - V characteristic curves of I_{sd} as a function of V_{sd} at different V_g after electromigration. V_g can modulate the range of current blockade region. ($T = 1.8$ K; the whole electrical transport measurements in device B were carried out at this cryogenic temperature). (b) The differential conductance dI/dV_{sd} as a function of V_g that is swept back and forth at zero bias voltage $V_{sd} = 0$ mV, which exhibits two set of Coulomb oscillation patterns. Traces have been offset vertically for clarity. (c, d) The two-dimensional dI_{sd}/dV_{sd} maps corresponding to the black line ($V_g = 2 - 14$ V) and red line ($V_g = -14 - 2$ V) in (b), respectively, plotted as a function of V_g and V_{sd} in device B. The Coulomb diamond, marked by the solid line, represents a single electron tunnelling through the central molecule. The intersections of these Coulomb edges at zero bias are charge degeneracy points or resonant tunneling points. The gate efficiency factor α is 0.0076 for (c) and 0.008 for (d). The energies of excitons marked by red arrows are approximately 20 meV in (c) and 27 meV in (d), respectively. The excited states can presumably be assigned

to the longitudinal vibration of the Dy ion coupled to the carbon cage [*Applied Physics Express* **13**, 105002 (2020).], rather than an intrinsic vibrational mode of the carbon cage.

Figure R3.6 | Zeeman effect of device B. (a, c) Coloured maps of the current for state 1 (a), and state 2 (c), as a function of an applied magnetic field and relative energy tuned by V_g (converting V_g to energy using the gate efficiency factor α) in device B. The bias voltage is fixed at 2 mV. The maps demonstrate clear evidence of Zeeman effect and a level splitting behavior of the $(N+1)$ charge state. The color bar represents the current intensity. (b, d) Addition energy and level splitting for the $(N+1)$ charge states as a function of magnetic field of state 1 (b) and state 2 (d). The addition energy defined as $E_{add}(N) = \mu_{N+1} - \mu_N = E_C + E_{N+1} - E_N$ (where μ is the chemical potential, E_C is the charging energy, E_{N+1} and E_N are energy level of the $(N+1)$ and N charge states) is calculated by measuring the spacing between consecutive Coulomb peaks. $\Delta\mu_{N+1}$ is

the energy level spacing between the $(N+1)$ and N charge states, and Δ_{N+1} is the energy level splitting of the $(N+1)$ charge state. We fit the data to obtain the effective g -factor. The g^* value for the level splitting Δ_{N+1} of the $(N+1)$ charge state reaches approximately 20, which is comparable to the splitting in the ground state $|\pm 15/2\rangle$ of Dy^{3+} . The extracted g^* value from the slope of $\Delta\mu_{N+1}$ is approximately -2 , indicating that the ground state of the N charge state with magnetic moment of $\sim 8 \mu_B$ may originate from either the excited state of Dy^{3+} or more likely a hybridized state with carbon cage.

Figure R3.7 | Evolution of peak amplitude and MR of the resonance tunneling point for the N charge state in device B. (a, c) The resonance tunneling peak of state 1 and state 2 for the N charge state ($V_g \sim -2$ V for state 1 and $V_g \sim -10$ V for state 2, indicated by black arrows in Fig. R3.5(c, d)) measured under different magnetic fields ($B = 0, 2, 4, 6, 8$ T) at a fixed bias voltage ($V_{sd} = 2$ mV). The presence of the magnetic field results in suppression of the peak amplitude. (b, d) Magnetic field dependence of MR extracted from peaks' amplitudes of the resonance tunneling point in (a, c) for state

1 (**b**) and state 2 (**d**). The MR of state 1 can reach 45% at $B = 9$ T, and that of state 2 is 40%. The N charge state exhibits a certain degree of hybridization behavior from the metal-cage hybrid state mechanism. There is small variation in the MR properties of the two molecular states.

Action: We added Figs. R3.5, R3.6, and R3.7 in Supplementary Information as new Fig. S1, S2, and S3.

In summary, we have fully addressed the reviewer's concerns that are extremely inspiring, and we believe that our revised manuscript has been significantly improved. We hope that our responses will meet the reviewer's expectations.

REVIEWERS' COMMENTS

Reviewer #1 (Remarks to the Author):

The reported report is very interesting.
However, the manuscript has to be improved in several points.

'N-1/N charge degeneracy', please specify what this is for the general audience.

The sentence 'The red line region corresponds to the modulation ability of our device, which is estimated to be $\Delta\mu = 1.3 \mu B$ for GN at approximately 3– 10 MV/cm, higher than that in most molecular crystals or single TiH molecules on MgO' is highly misleading as well as Figure 2(h). I think this figure has to be removed. For MnPhOMe, the value is even reported in the wrong place. Moreover, the result of the present work cannot be extrapolated to zero electric field, and therefore they cannot be properly compared with the molecular systems, as here the reported system needs to be activated by a rather large electric field.

What is the direction of the magnetic field with respect to the electric field? Is this relevant for the observed behaviour?

MR acronym is used before its definition in 'Large MR and potential multistate data storage In this section, we demonstrate the disparity in MR'.

Please clarify to which factors the Author refer to in 'Additional factors associated with electric dipole moments have recently been employed for spin–electric control'.

In the sentence 'Our DFT calculations demonstrated that the HOMO, LUMO, and their 201 adjacent molecular orbitals likely originate from either the 4f orbitals of Dy³⁺ or the metal-cage hybrid orbitals (Fig. 4(a, c) and Supplementary Figs. S5, S6, and Table S1).' please change the word demonstrate with a more appropriate one as after it is said 'likely originated'.

In the sentence, 'The relative energies of the ground and excited states extracted from the peaks of the d²I_{sd}/dV²_{sd} curves in Fig. 2(b, e) are shown in Fig. 2(c, f)' please clarify how these relative energies have been extracted.

Reviewer #2 (Remarks to the Author):

I thank and congratulate the authors for their work. My concerns have been abundantly addressed, and it seems that this is also the case for the concerns of the other reviewers. I do agree that the revised manuscript has been significantly improved as a result. I recommend publication.

Reviewer #3 (Remarks to the Author):

The authors have addressed all my comments and questions. Now I recommend its publication.

RESPONSE TO REVIEWERS' COMMENTS (NCOMMS-23-50571A)

We sincerely appreciate the thorough evaluation of our work by all Reviewers and Editors and the time they spent on our manuscript. In the revised manuscript and Response Letter, we have essentially addressed all the questions and comments raised by the Reviewers. Below, we provide a detailed point-by-point response (marked in blue) to each comment. The text changes are highlighted red in the revised manuscript.

Reviewer #1 (Remarks to the Author)

Comments:

The reported report is very interesting.

However, the manuscript has to be improved in several points.

Reply: We highly appreciate the reviewer for the comments on the novelty and significance of our work. We have revised the ambiguous or incorrect contents in the manuscript according to the suggestions of the reviewer. We provide the detailed point-by-point response to reviewer's concerns below.

(1) 'N-1/N charge degeneracy', please specify what this is for the general audience.

Reply: We thank the reviewer for this important question. The charge degeneracy points refer to the electron tunneling when the chemical potential of the molecule aligns with the Fermi level of both source and drain electrodes, which is represented by the crossing of the Coulomb diamond edges at zero bias in the Coulomb stability diagram, also known as resonant tunneling points [*Nature* **423**, 422-425 (2003); *Nature communications* **7**, 11595 (2016); *Nature Communications* **14**, 7486 (2023)]. The $N-1$ and N represent the number of electrons on the molecule. The charge degeneracy point in the manuscript represents the transition between $N-1$ and N charge states.

Action: We have revised the description of the charge degeneracy point in the

manuscript and added corresponding explanations.

(2) The sentence ‘The red line region corresponds to the modulation ability of our device, which is estimated to be $\Delta\mu = 1.3 \mu_B$ for GN at approximately 3–10 MV/cm, higher than that in most molecular crystals or single TiH molecules on MgO’ is highly misleading as well as Figure 2(h). I think this figure has to be removed. For MnPhOMe, the value is even reported in the wrong place. Moreover, the result of the present work cannot be extrapolated to zero electric field, and therefore they cannot be properly compared with the molecular systems, as here the reported system needs to be activated by a rather large electric field.

Reply: We thank the reviewer for raising this insightful question and nice suggestion. In the experiment, the molecular state transition was modulated by an electric field, thereby changing its magnetic moment. The effective magnetic moment of ground state G_N has been changed by $1.3 \mu_B$ under an electric field of 3–10 MV/cm. In a broad sense, this is an electric-field manipulation of magnetism phenomenon. Therefore, we compare the modification to the effective magnetic moment under an electric field with other systems. But now, we have seriously considered and agree with the viewpoint of the reviewer that this comparison is misleading. This is primarily due to the requirement of a threshold electric field for modulating the magnetic moment of Dy in our work, which cannot be extrapolated to lower or zero fields. Consequently, following the reviewer's suggestion, we deleted this inappropriate comparison, including Fig. 2(h) and the corresponding description in the manuscript.

Action: We have deleted the inappropriate comparison along with Fig. 2(h) and the corresponding description in the manuscript.

(3) What is the direction of the magnetic field with respect to the electric field? Is this relevant for the observed behaviour?

Reply: We thank the reviewer for this important question. The direction of the magnetic field applied in our experiment is perpendicular to the direction of the current. The

devices are placed on a horizontal plane, and the direction of the applied magnetic field is vertical. Whether the observed magnetic behaviour is relevant with the direction of the magnetic field depends on whether Dy@C₈₄ molecules have a large magnetic anisotropy. Dy@C₈₂ has shown paramagnetic properties even at 1.8 K [*The Journal of Physical Chemistry B* **104**, 1473-1482 (2000)]. Large magnetic anisotropy is unlikely to exist in Dy@C₈₄ akin to that in Dy@C₈₂. Theoretical calculations results reveal an energy difference of ~500 μeV between magnetic moments oriented along two orthogonal directions, indicating that Dy@C₈₄ is more likely to exhibit paramagnetic behaviour. In this case, the survival of the magnetic state protected by uniaxial magnetic anisotropy becomes challenging, as does the manifestation of anisotropic magnetoresistance behaviour upon reorientation of the magnetic moment.

To summarize, Dy@C₈₄ is more likely to exhibit paramagnetic behavior, and the observed magnetic behaviour is irrelevant to the direction of the magnetic field. The magnetism of the Dy atom changes with the transition between the two molecular states. Therefore, the switching of molecular states and single-atom magnetism can be effectively controlled by an electric field, which manifests alterations in the effective magnetic moment and MR behaviour.

Action: We have added the description about the direction of the magnetic field with respect to the electric field in the manuscript.

(4) MR acronym is used before its definition in ‘Large MR and potential multistate data storage

In this section, we demonstrate the disparity in MR’.

Reply: We thank the reviewers for pointing out the lack of rigor in our usage of the MR acronym before its definition. The definition of MR acronym is added in the text of the manuscript.

Action: We have revised the definition of MR acronym in the manuscript.

(5) Please clarify to which factors the Author refer to in ‘Additional factors associated with electric dipole moments have recently been employed for spin–electric control’.

Reply: We thank the reviewer for raising this important question and nice suggestion. Additional factors include broken symmetries [*Physical review letters* **101**, 217201 (2008); *Nature Physics* **17**, 1205-1209 (2021).] and electrical modulation of exchange interactions [*Physical Review B* **82**, 045429 (2010); *Nature Materials* **18**, 329-334 (2019).], etc.

Action: We have clarified additional factors such as electrical modulation of exchange interactions or broken symmetries in the manuscript.

(6) In the sentence ‘Our DFT calculations demonstrated that the HOMO, LUMO, and their 201 adjacent molecular orbitals likely originate from either the 4f orbitals of Dy³⁺ or the metal-cage hybrid orbitals (Fig. 4(a, c) and Supplementary Figs. S5, S6, and Table S1).’ please change the word demonstrate with a more appropriate one as after it is said ‘likely originated’.

Reply: We are grateful to the reviewer to point out our inappropriate word. We want to change the word ‘demonstrate’ to ‘show’.

Action: We have revised the word ‘demonstrate’ to ‘show’ in the manuscript.

(7) In the sentence, ‘The relative energies of the ground and excited states extracted from the peaks of the d^2I/dV^2 curves in Fig. 2(b, e) are shown in Fig. 2(c, f)’ please clarify how these relative energies have been extracted.

Reply: We thank the reviewer for this important question. The data points in Fig. 2(c, f) are extracted from the peaks’ position of the d^2I/dV^2 curves at different V_g in Fig. 2(b, e) and convert the voltages represented by the positions into energies (eV). We set the energy at zero bias to zero, and we get the relative energies of the ground and excited states. The d^2I/dV^2 map provides an enhanced visualization to illustrate the Zeeman shift for each state, which were also used in previous works [*Nature* **468**, 1084-1087

(2010); *Nano letters* **16**, 7509-7513 (2016); *ACS nano* **11**, 5879-5883 (2017)]. Subsequently, curve fitting is employed to determine the effective g -factors for different states. The data points of state 1 recorded before the 1.5 T exhibit significant variability in their behavior. Therefore, these “bad” data points are excluded during the fitting process.

Reviewer #2 (Remarks to the Author)

Comments:

I thank and congratulate the authors for their work. My concerns have been abundantly addressed, and it seems that this is also the case for the concerns of the other reviewers. I do agree that the revised manuscript has been significantly improved as a result. I recommend publication.

Reply: We highly appreciate the reviewer for high evaluations on the significance of our work and insightful comments that helped us improve the quality of our work.

Reviewer #3 (Remarks to the Author)

Comments:

The authors have addressed all my comments and questions. Now I recommend its publication.

Reply: We are highly grateful to the reviewer for his/her positive comments on our work. The reviewer's insightful views and evaluations have been a great help to our work.